# MOCODA: Model-based Counterfactual Data Augmentation

**Silviu Pitis**[*1]  **Elliot Creager**[1]  **Ajay Mandlekar**[2]  **Animesh Garg**[1,2]
[1]University of Toronto and Vector Institute, [2]NVIDIA

## Abstract

The number of states in a dynamic process is exponential in the number of objects, making reinforcement learning (RL) difficult in complex, multi-object domains. For agents to scale to the real world, they will need to react to and reason about unseen combinations of objects. We argue that the ability to recognize and use local factorization in transition dynamics is a key element in unlocking the power of multi-object reasoning. To this end, we show that (1) known local structure in the environment transitions is sufficient for an exponential reduction in the sample complexity of training a dynamics model, and (2) a locally factored dynamics model provably generalizes out-of-distribution to unseen states and actions. Knowing the local structure also allows us to predict *which* unseen states and actions this dynamics model will generalize to. We propose to leverage these observations in a novel Model-based Counterfactual Data Augmentation (MOCODA) framework. MOCODA applies a learned locally factored dynamics model to an augmented distribution of states and actions to generate counterfactual transitions for RL. MOCODA works with a broader set of local structures than prior work and allows for direct control over the augmented training distribution. We show that MOCODA enables RL agents to learn policies that generalize to unseen states and actions. We use MOCODA to train an offline RL agent to solve an out-of-distribution robotics manipulation task on which standard offline RL algorithms fail.[1]

## 1 Introduction

Modern reinforcement learning (RL) algorithms have demonstrated remarkable success in several different domains such as games [42, 53] and robotic manipulation [23, 4]. By repeatedly attempting a single task through trial-and-error, these algorithms can learn to collect useful experience and eventually solve the task of interest. However, designing agents that can generalize in *off-task* and *multi-task* settings remains an open and challenging research question. This is especially true in the offline and zero-shot settings, in which the training data might be unrelated to the target task, and may lack sufficient coverage over possible states.

One way to enable generalization in such cases is through structured representations of states, transition dynamics, or task spaces. These representations can be directly learned, sourced from known or learned abstractions over the state space, or derived from causal knowledge of the world. Symmetries present in such representations enable compositional generalization to new configurations of states or tasks, either by building the structure into the function approximator or algorithm [28, 58, 15, 43], or by using the structure for data augmentation [3, 33, 51].

In this paper, we extend past work on structure-driven data augmentation by using a locally factored model of the transition dynamics to generate counterfactual training distributions. This enables agents to generalize beyond the support of their original training distribution, including to novel

---

[*]Correspondence to `spitis@cs.toronto.edu`
[1]Visualizations & code available at `https://sites.google.com/view/mocoda-neurips-22/`

36th Conference on Neural Information Processing Systems (NeurIPS 2022).

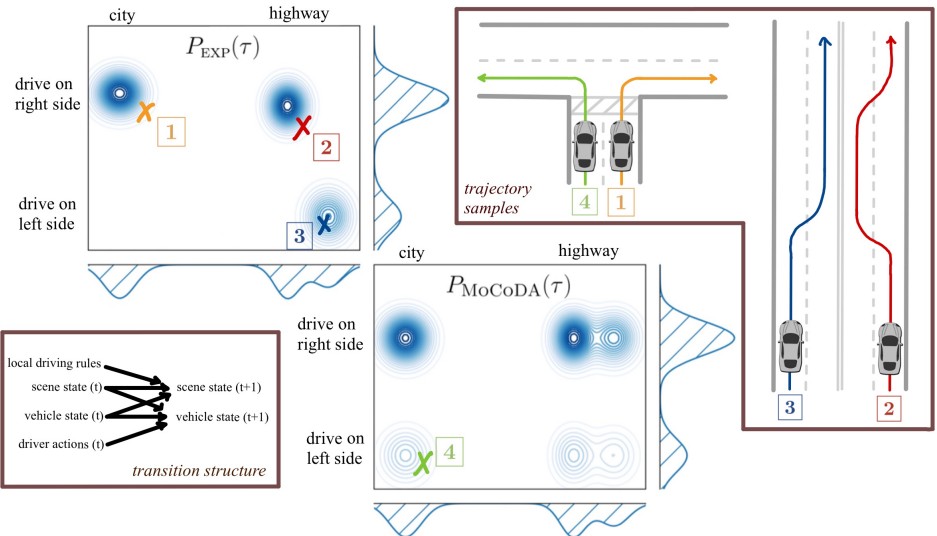

Figure 1: **Out-of-Distribution Generalization using MoCoDA**: A US driver can use MoCoDA to quickly adapt to driving in the left lane during a UK trip. Their prior experience $P_{\text{EMP}}(\tau)$ (**top left**) contains mostly right-driving experience (e.g. 1, 2) and a limited amount of left-driving experience after renting the car in the UK (e.g. 3). A locally factored model that captures the transition structure (**bottom left**) allows the agent to accurately sample counterfactual experience from $P_{\text{MoCoDA}}(\tau)$ (**bottom center**), including novel left-lane city driving maneuvers (e.g. 4). This enables fast adaptation when learning an optimal policy for the new task (UK driving). Our framework MoCoDA draws single-step transition samples from $P_{\text{MoCoDA}}(\tau)$ given $P_{\text{EMP}}(\tau)$ and knowledge of the causal structure; several realizations of this framework are described in Section 4.

tasks where learning the optimal policy requires access to states never seen in the experience buffer. Our key insight is that a learned dynamics model that accurately captures local causal structure (a "locally factored" dynamics model) will predictably exhibit good generalization performance outside the empirical training distribution. We propose Model-based Counterfactual Data Augmentation (MoCoDA), which generates an augmented state-action distribution where its locally factored dynamics model is likely to perform well, then applies its dynamics model to generate new transition data. By training the agent's policy and value modules on this augmented dataset, they too learn to generalize well out-of-distribution. To ground this in an example, we consider how a US driver might use MoCoDA to adapt to driving on the left side of the road while on vacation in the UK (Figure 1). Given knowledge of the target task, we can even focus the augmented distribution on relevant areas of the state-action space (e.g., states with the car on the left side of the road).

Our main contributions are:

A. Our proposed method, MoCoDA, leverages a masked dynamics model for data-augmentation in locally-factored settings, which relaxes strong assumptions made by prior work on factored MDPs and counterfactual data augmentation.

B. MoCoDA allows for direct control of the state-action distribution on which the agent trains; we show that controlling this distribution in a task relevant way can lead to improved performance.

C. We demonstrate "zero-shot" generalization of a policy trained with MoCoDA to states that the agent has never seen. With MoCoDA, we train an offline RL agent to solve an out-of-distribution robotics manipulation task on which standard offline RL algorithms fail.

## 2 Preliminaries

### 2.1 Background

We model the environment as an infinite-horizon, reward-free Markov Decision Process (MDP), described by tuple $\langle \mathcal{S}, \mathcal{A}, P, \gamma \rangle$ consisting of the state space, action space, transition function, and discount factor, respectively [52, 57]. We use lowercase for generic instances and uppercase for

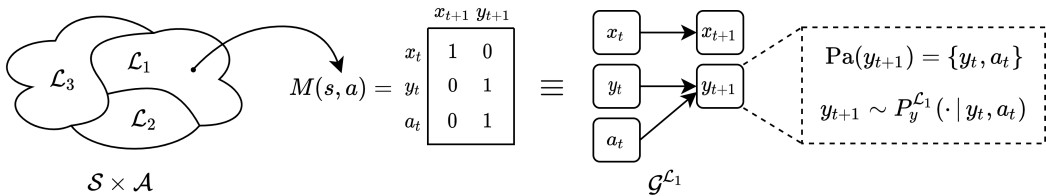

Figure 2: **Locally Factored Dynamics:** The state-action space $\mathcal{S} \times \mathcal{A}$ is divided into local subsets, $\mathcal{L}_1, \mathcal{L}_2, \mathcal{L}_3$, which each have their own factored causal structure, $\mathcal{G}^{\mathcal{L}}$. The local transition model $P^{\mathcal{L}}$ is factored according to $\mathcal{G}^{\mathcal{L}}$; e.g., in the example shown, $P^{\mathcal{L}}(x_t, y_t, a_t) = [P_x(x_t), P_y(y_t, a_t)]$.

variables (e.g., $s \in \text{range}(S) \subseteq \mathcal{S}$, though we also abuse notation and write $S \in \mathcal{S}$). A *task* is defined as a tuple $\langle r, P_0 \rangle$, where $r : \mathcal{S} \times \mathcal{A} \to \mathbb{R}$ is a reward function and $P_0$ is an initial distribution over $S$. The goal of the agent given a task is to learn a policy $\pi : \mathcal{S} \to \mathcal{A}$ that maximizes value $\mathbb{E}_{P,\pi} \sum_t \gamma^t r(s_t, a_t)$. Model-based RL is one approach to solving this problem, in which the agent learns a model $P_\phi$ of the transition dynamics $P$. The model is "rolled out" to generate "imagined" trajectories, which are used either for direct planning [11, 8], or as training data for the agent's policy and value functions [56, 20].

**Factored MDPs**. A factored MDP (FMDP) is a type of MDP that assumes a globally factored transition model, which can be used to exponentially improve the sample complexity of RL [16, 24, 45]. In an FMDP, states and actions are described by a set of variables $\{X^i\}$, so that $\mathcal{S} \times \mathcal{A} = \mathcal{X}^1 \times \mathcal{X}^2 \times \ldots \times \mathcal{X}^n$, and each state variable $X^i \in \mathcal{X}^i$ ($\mathcal{X}^i$ is a subspace of $\mathcal{S}$) is dependent on a subset of state-action variables (its "parents" $\text{Pa}(X^i)$) at the prior timestep, $X^i \sim P_i(\text{Pa}(X^i))$. We call a set $\{X^j\}$ of state-action variables a "parent set" if there exists a state variable $X^i$ such that $\{X^j\} = \text{Pa}(X^i)$. We say that $X^i$ is a "child" of its parent set $\text{Pa}(X^i)$. We refer to the tuple $\langle X^i, \text{Pa}(X^i), P_i(\cdot) \rangle$ as a "causal mechanism".

**Local Causal Models**. Because the strict global factorization assumed by FMDPs is rare, recent work on data augmentation for RL and object-oriented RL suggests that transition dynamics might be better understood in a local sense, where all objects may interact with each other over time, but in a locally sparse manner [15, 28, 39]. Our work uses an abridged version of the Local Causal Model (LCM) framework [51], as follows: We assume the state-action space decomposes into a disjoint union of local neighborhoods: $\mathcal{S} \times \mathcal{A} = \mathcal{L}_1 \sqcup \mathcal{L}_2 \sqcup \cdots \sqcup \mathcal{L}_n$. A neighborhood $\mathcal{L}$ is associated with its own transition function $P^{\mathcal{L}}$, which is factored according to its graphical model $\mathcal{G}^{\mathcal{L}}$ [29]. We assume no two graphical models share the same structure[2] (i.e., the structure of $\mathcal{G}^{\mathcal{L}}$ uniquely identifies $\mathcal{L}$). Then, analogously to FMDPs, if $(s_t, a_t) \in \mathcal{L}$, each state variable $X^i_{t+1}$ at the next time step is dependent on its parents $\text{Pa}^{\mathcal{L}}(X^i_{t+1})$ at the prior timestep, $X^i_{t+1} \sim P_i^{\mathcal{L}}(\text{Pa}^{\mathcal{L}}(X^i_{t+1}))$. We define mask function $M : \mathcal{S} \times \mathcal{A} \to \{\mathcal{L}_i\}$ that maps $(s, a) \in \mathcal{L}$ to the adjacency matrix of $\mathcal{G}^{\mathcal{L}}$. This formalism is summarized in Figure 2, and differs from FMDPs in that each $\mathcal{L}$ has its own factorization.

Given knowledge of $M$, the Counterfactual Data Augmentation (CoDA) framework [51] allowed agents to stitch together empirical samples from disconnected causal mechanisms to derive novel transitions. It did this by swapping compatible components between the observed transitions to create new ones, arguing that this procedure can generate exponentially more data samples as the number of disconnected causal components grows. CoDA was shown to significantly improve sample complexity in several settings, including the offline RL setting and a goal-conditioned robotics control setting. Because CoDA relied on empirical samples of the causal mechanisms to generate data in a model-free fashion, however, it required that the causal mechanisms be completely disentangled. The proposed MOCODA *leverages a dynamics model to improve upon model-free CoDA* in several respects: (a) by using a learned dynamics model, MOCODA works with overlapping parent sets, (b) by explicitly modeling the parent distribution, MOCODA allows the agent to control the overall data distribution, (c) MOCODA demonstrates zero-shot generalization to new areas of the state space, allowing the agent to solve tasks that are entirely outside the original data distribution.

---

[2]This assumption is a matter of convenience that makes counting local subspaces in Section 3 slightly easier and simplifies our implementation of the locally factored dynamics model in Section 4. To accommodate cases where subspaces with different dynamics share the same causal structure, one could identify local subspaces using a latent variable rather than the mask itself, which we leave for future work.

## 2.2 Related Work

**RL with Structured Dynamics**. A growing literature recognizes the advantages that structure can provide in RL, including both improved sample efficiency [37, 5, 19] and generalization performance [62, 59, 54]. Some of these works involve sparse interactions whose structure changes over time [15, 28], which is similar to and inspires the locally factored setup assumed by this paper. Most existing work focuses on leveraging structure to improve the architecture and generalization capability of the function approximator [62]. Although MoCoDA also uses the structure for purposes of improving the dynamics model, our proposed method is among the few existing works that also use the structure for data augmentation [38, 40, 51].

Several past and concurrent works aim to tackle unsupervised object detection [36, 12] (i.e., learning an entity-oriented representation of states, which is a prerequisite for learning the dynamics factorization) and learning the dynamics factorization [27, 60]. These are both open problems that run orthogonal to MoCoDA. We expect that as solutions for unsupervised object detection and factored dynamics discovery improve, MoCoDA will find broader applicability.

**RL with Causal Dynamics**. Adopting this formalism allows one to cast several important problems within RL as questions of causal inference, such as off-policy evaluation [7, 44], learning baselines for model-free RL [41], and policy transfer [25]. Lu et al. [38] applied SCM dynamics to data augmentation in continuous sample spaces, and discussed the conditions under which the generated transitions are uniquely identifiable counterfactual samples. This approach models state and action variables as unstructured vectors, emphasizing benefit in modeling action interventions for settings such as clinical healthcare where exploratory policies cannot be directly deployed. We take a complementary approach by modeling structure *within* state and action variables, and our augmentation scheme involves sampling entire causal mechanisms (over multiple state or action dimensions) rather than action vectors only. See Appendix F for a more detailed discussion of how MoCoDA sampling relates to causal inference and counterfactual reasoning.

## 3 Generalization Properties of Locally Factored Models

### 3.1 Sample Complexity of Training a Locally Factored Dynamics Model

In this subsection, we provide an original adaptation of an elementary result from model-based RL to the *locally* factored setting, to show that factorization can exponentially improve sample complexity. We note that several theoretical works have shown that the FMDP structure can be exploited to obtain similarly strong sample complexity bounds in the FMDP setting. Our goal here is not to improve upon these results, but to adapt a small part (model-based generalization) to the significantly more general locally factored setting and show that local factorization is enough for (1) *exponential gains in sample complexity* and (2) *out-of-distribution generalization* with respect to the empirical joint, to a set of states and actions that may be exponentially larger than the empirical set. Note that the following discussion applies to tabular RL, but we apply our method to continuous domains.

**Notation**. We work with finite state and action spaces ($|\mathcal{S}|, |\mathcal{A}| < \infty$) and assume that there are $m$ local subspaces $\mathcal{L}$ of size $|\mathcal{L}|$, such that $m|\mathcal{L}| = |\mathcal{S}||\mathcal{A}|$. For each subspace $\mathcal{L}$, we assume transitions factor into $k$ causal mechanisms $\{P_i\}$, each with the same number of possible children, $|c_i|$, and the same number of possible parents, $|\text{Pa}_i|$. Note $m\Pi_i|c_i| = |\mathcal{S}|$ (child sets are mutually exclusive) but $m\Pi_i|\text{Pa}_i| \geq |\mathcal{S}||\mathcal{A}|$ (parent sets may overlap).

**Theorem 1.** *Let $n$ be the number of empirical samples used to train the model of each local causal mechanism, $P_{i,\theta}^{\mathcal{L}}$ at each configuration of parents $Pa_i = x$. There exists constant $c$ such that, if*

$$n \geq \frac{ck^2|c_i|\log(|\mathcal{S}||\mathcal{A}|/\delta)}{\epsilon^2},$$

*then, with probability at least $1 - \delta$, we have:*

$$\max_{(s,a)} \|P(s,a) - P_\theta(s,a)\|_1 \leq \epsilon.$$

*Sketch of Proof.* We apply a concentration inequality to bound the $\ell_1$ error for fixed parents and extend this to a bound on the $\ell_1$ error for a fixed $(s,a)$ pair. The conclusion follows by a union bound across all states and actions. See Appendix A for details. □

To compare to full-state dynamics modeling, we can translate the sample complexity from the per-parent count $n$ to a total count $N$. Recall $m\Pi_i|c_i| = |\mathcal{S}|$, so that $|c_i| = (|\mathcal{S}|/m)^{1/k}$, and $m\Pi_i|\text{Pa}_i| \geq |\mathcal{S}||\mathcal{A}|$. We assume a small constant overlap factor $v \geq 1$, so that $|\text{Pa}_i| = v(|\mathcal{S}||\mathcal{A}|/m)^{1/k}$. We need the total number of component visits to be $n|\text{Pa}_i|km$, for a total of $nv(|\mathcal{S}||\mathcal{A}|/m)^{1/k}m$ state-action visits, assuming that parent set visits are allocated evenly, and noting that each state-action visit provides $k$ parent set visits. This gives:

**Corollary 1.** *To bound the error as above, we need to have*

$$N \geq \frac{cmk^2(|\mathcal{S}|^2|\mathcal{A}|/m^2)^{1/k}\log(|\mathcal{S}||\mathcal{A}|/\delta)}{\epsilon^2},$$

*total train samples, where we have absorbed the overlap factor $v$ into constant c.*

Comparing this to the analogous bound for full-state model learning (Agarwal et al. [1], Prop. 2.1):

$$N \geq \frac{c|\mathcal{S}|^2|\mathcal{A}|\log(|\mathcal{S}||\mathcal{A}|/\delta)}{\epsilon^2},$$

we see that we have gone from super-linear $O(|\mathcal{S}|^2|\mathcal{A}|\log(|\mathcal{S}||\mathcal{A}|))$ sample complexity in terms of $|S||A|$, to the exponentially smaller $O(mk^2(|\mathcal{S}|^2|\mathcal{A}|/m^2)^{1/k}\log(|\mathcal{S}||\mathcal{A}|))$.

This result implies that *for large enough $|\mathcal{S}||\mathcal{A}|$ our model must generalize to unseen states and actions*, since the number of samples needed ($N$) is exponentially smaller than the size of the state-action space ($|\mathcal{S}||\mathcal{A}|$). In contrast, if it did not, then sample complexity would be $\Omega(|\mathcal{S}||\mathcal{A}|)$.

**Remark 3.1.** *The global factorization property of FMDPs is a strict assumption that rarely holds in reality. Although local factorization is broadly applicable and significantly more realistic than the FMDP setting, it is not without cost. In FMDPs, we have a single subspace ($m = 1$). In the locally factored case, the number of subspaces $m$ is likely to grow exponentially with the number of factors $k$, as there are exponentially many ways that $k$ factors can interact. To be more precise, there are $k2^k$ possible bipartite graphs from $k$ nodes to $k$ nodes. Nevertheless, by comparing bases ($2 \ll |\mathcal{S}||\mathcal{A}|$), we see that we still obtain exponential gains in sample complexity from the locally factored approach.*

### 3.2 Training Value Functions and Policies for Out-of-Distribution Generalization

In the previous subsection, we saw that a locally factored dynamics model provably generalizes outside of the empirical joint distribution. A natural question is whether such *local factorization can be leveraged to obtain similar results for value functions and policies*?

We will show that the answer is *yes*, but perhaps counter-intuitively, it is not achieved by directly training the value function and policy on the empirical distribution, as is the case for the dynamics model. The difference arises because learned value functions, and consequently learned policies, involve the long horizon prediction $\mathbb{E}_{P,\pi}\sum_{t=0}^{\infty}\gamma^t r(s_t, a_t)$, which may not benefit from the local sparsity of $\mathcal{G}^{\mathcal{L}}$. When compounded over time, sparse local structures can quickly produce an entangled long horizon structure (cf. the "butterfly effect"). Intuitively, even if several pool balls are far apart and locally disentangled, future collisions are central to planning and the optimal policy depends on the relative positions of all balls. This applies even if rewards are factored (e.g., rewards in most pool variants) [54].

We note that, although temporal entanglement may be exponential in the branching factor of the unrolled causal graph, it's possible for the long horizon structure to stay sparse (e.g., $k$ independent factors that never interact, or long-horizon disentanglement between descision relevant and decision irrelevant variables [19]). It's also possible that other regularities in the data will allow for good out-of-distribution generalization. Thus, we cannot claim that value functions and policies will never generalize well out-of-distribution (see Veerapaneni et al. [58] for an example when they do). Nevertheless, we hypothesize that exponentially fast entanglement does occur in complex natural systems, making direct generalization of long horizon predictions difficult.

Out-of-distribution generalization of the policy and value function can be achieved, however, by leveraging the generalization properties of a locally factored dynamics model. We propose to do this by generating out-of-distribution states and actions (the "augmented parent distribution"), and then applying our learned dynamics model to generate transitions that are used to train the policy and value function. We call this process Model-based Counterfactual Data Augmentation (MoCoDA).

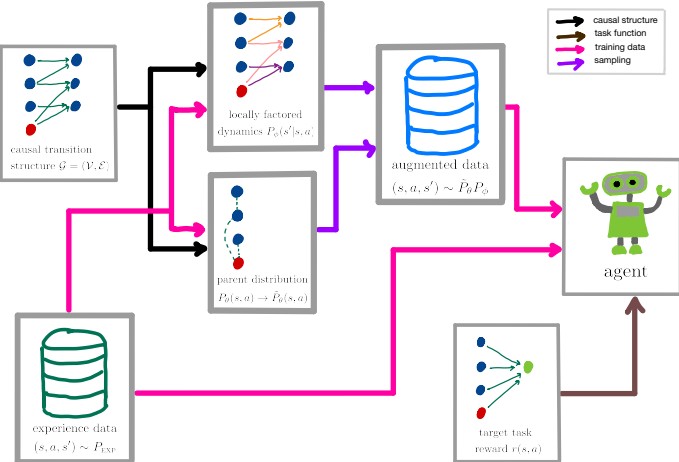

Figure 3: **RL training with MoCoDA**: We use the empirical dataset to train parent distribution model $P_\theta(s, a)$ and locally factored dynamics model $P_\phi(s' \mid s, a)$, both informed by the local structure. The dynamics model is applied to the augmented parent distribution $\tilde{P}_\theta(s, a)$ to produce augmented dataset $\tilde{P}_\theta P_\phi$. The augmented & empirical datasets are labeled with the target task reward, $r(s, a)$ and fed into the RL algorithm as training data.

## 4 Model-based Counterfactual Data Augmentation

In the previous section, we discussed how locally factored dynamics model can generalize beyond the empirical dataset to provide accurate predictions on an augmented state-action distribution we call the "parent distribution". We now seek to leverage this out-of-distribution generalization in the dynamics model to bootstrap the training of an RL agent. Our approach is to control the agent's training distribution $P(s, a, s')$ via the locally factored dynamics $P_\phi(s'|s, a)$ and the parent distribution $P_\theta(s, a)$ (both trained using experience data). This allows us to sample *augmented* transitions (perhaps unseen in the experience data) for consumption by a downstream RL agent. We call this framework MoCoDA, and summarize it using the following three-step process:

**Step 1**  Given known parent sets, model the parent distribution $P_\theta(s, a)$ and generate an appropriate augmented parent distribution $\tilde{P}_\theta(s, a)$.

**Step 2**  Apply a learned dynamics model $P_\phi(s'|s, a)$ to augmented parent distribution to generate "augmented dataset" of transitions $(s, a, s')$.

**Step 3**  Use augmented dataset $s, a, s' \sim \tilde{P}_\theta P_\phi$ (alongside experience data, if desired) to train an off-policy RL agent on the (perhaps novel) target task.

Figure 3 illustrates this framework in a block diagram. An instance of MoCoDA is realized by specific choices at each step. For example, the original CoDA method [51] is an instance of MoCoDA, which (1) generates the augmented parent distribution by uniformly swapping non-overlapping parent sets, and (2) uses subsamples of empirical transitions as a locally factored dynamics model. CoDA works when local graphs have non-overlapping parent sets, but it does not allow for control over the parent distribution and does not work in cases where parent sets overlap. MoCoDA generalizes CoDA, alleviating these restrictions and allowing for significantly more design choices.

### 4.1  Augmenting the Parent Distribution

How should the parent distribution be augmented (Step 1) to generate the augmented dataset? In other words, after fitting $P_\theta(s, a)$ to experience, how should we realize $\tilde{P}_\theta(s, a)$? We describe some options below, noting that our proposals (MoCoDA, MoCoDA-U, MoCoDA-P) rely on knowledge of (possibly local) parent sets—i.e., they require the state to be decomposed into objects.

**Baseline Distributions.**  If we restrict ourselves to states and actions in the empirical dataset (**Emp**) or short-horizon rollouts that start in the empirical state-action distribution (**Dyna**), as is typical in Dyna-style approaches [57, 20], we limit ourselves to a small neighborhood of the empirical state-action distribution. This forgoes the opportunity to train our off-policy RL agent on out-of-distribution data that may be necessary for learning the target task.

Another option is to sample random state-actions from $\mathcal{S} \times \mathcal{A}$ (**RAND**). While this provides coverage of all $(s, a)$ relevant to the target task, there is no guarantee that our locally factorized model generalizes well in RAND. The proof of Theorem 1 shows that our model only generalizes well to a particular $(s, a)$ if each component generalizes well on the configurations of each parent set in that $(s, a)$. In context of Theorem 1, this occurs only if the empirical data used to train our model contained at least $n$ samples for each set of parents in $(s, a)$. This suggests focusing on data whose parent sets have sufficient support in the empirical dataset.

**The MOCODA distribution.** We do this by constraining the marginal distribution of each parent set (within local neighborhood $\mathcal{L}$) in the augmented distribution to match the corresponding marginal in the empirical dataset. As there are many such distributions, in absence of additional information, it is sensible to choose the one with maximum entropy [21]. We call this maximum entropy, marginal matching distribution the **MOCODA** augmented distribution. Figure 1 provides an illustrative example of going from EMP (driving primarily on the right side) to MOCODA (driving on both right and left). We propose an efficient way to generate the MOCODA distribution using a set of Gaussian Mixture Models, one for each parent set distribution. We sample parent sets one at a time, conditioning on any previous partial samples due to overlap between parent sets. This process is detailed in Appendix B.

**Weaknesses of the MOCODA distribution.** Although our locally factored dynamics model is likely to generalize well on MOCODA, there are a few reasons why training our RL agent on MOCODA in Step 3 may yield poor results. First, if there are empirical imbalances within parent sets (some parent configurations more common than others), these imbalances will appear in MOCODA. Moreover, multiple such imbalances will compound exponentially, so that $(s, a)$ tuples with rare parent combinations will be extremely rare in MOCODA, even if the model generalizes well to them. Second, Support(MOCODA) may be so large that it makes training the RL algorithm in Step 3 inefficient. Finally, the cost function used in RL algorithms is typically an expectation over the training distribution, and optimizing the agent in irrelevant areas of the state-action space may hurt performance. The above limitations suggest that rebalancing MOCODA might improve results.

**MOCODA-U and MOCODA-P.** To mitigate the first weakness of MOCODA we might skew MOCODA toward the uniform distribution over its support, $\mathcal{U}(\text{Support}(\text{MOCODA}))$. Although this is possible to implement using rejection sampling when $k$ is small, exponential imbalance makes it impractical when $k$ is large. A more efficient implementation reweights the GMM components used in our MOCODA sampler. We call this approach (regardless of implementation) **MOCODA-U**. To mitigate the second and third weaknesses of MOCODA, we need additional knowledge about the target task—e.g., domain knowledge or expert trajectories. We can use such information to define a prioritized parent distribution **MOCODA-P** with support in Support(MOCODA), which can also be obtained via rejection sampling (perhaps on MOCODA-U to also relieve the initial imbalance).

## 4.2 The Choice of Dynamics Model and RL Algorithm

Once we have an augmented parent distribution, $\tilde{P}_\theta(s, a)$, we generate our augmented dataset by applying dynamics model $P_\phi(s' \mid s, a)$. The natural choice in light of the discussion in Section 3 is a locally factored model. This requires knowledge of the local factorization, which is more involved than the parent set knowledge used to generate the MOCODA distribution and its reweighted variants. We note, however, that a locally factored model may not be strictly necessary for MOCODA, so long as the underlying dynamics are factored. Although unfactored models do not perform well in our experiments, we hypothesize that a good model with enough in-distribution data and the right regularization might learn to implicitly respect the local factorization. The choice of model architecture is not core to our work, and we leave exploration of this possibility to future work.

**Masked Dynamics Model.** In our experiments, we assume access to a mask function $M : \mathcal{S} \times \mathcal{A} \rightarrow \{0, 1\}^{(|\mathcal{S}|+|\mathcal{A}|) \times |\mathcal{S}|}$ (perhaps learned [27, 51]), which maps states and actions to the adjacency map of the local graph $\mathcal{G}^\mathcal{L}$. Given this mask function, we design a dynamics model $P_\phi$ that accepts $M(s, a)$ as an additional input and respects the causal relations in the mask (i.e., mutual information $I(X_t^i; X_{t+1}^j \mid (S_t, A_t) \setminus X_t^i) = 0$ if $M(s_t, a_t)_{ij} = 0$). There are many architectures that enforce this constraint. In our experiments we opt for a simple one, which first embeds each of the $k$ parent sets: $f = [f_i(\text{Pa}_i)]_{i=1}^k$, and then computes the $j$-th child as a function of the sum of the masked embeddings, $g_j(M(s, a)_{\cdot,j} \cdot f)$. See Appendix B for further implementation details.

**The RL Algorithm.** After generating an augmented dataset by applying our dynamics model to the augmented distribution, we label the data with our target task reward and use the result to train an RL

agent. MOCODA works with a wide range of algorithms, and the choice of algorithm will depend on the task setting. For example, our experiments are done in an offline setup, where the agent is given a buffer of empirical data, with no opportunity to explore. For this reason, it makes sense to use offline RL algorithms, as this setting has proven challenging for standard online algorithms [34].

**Remark 4.1.** *The rationales for (1) regularizing the policy toward the empirical distribution in offline RL algorithms, and (2) training on the MOCODA distribution, are compatible: in each case, we want to restrict ourselves to state-actions where our models generalize well. By using MOCODA we expand this set beyond the empirical distribution. Thus, when we apply offline RL algorithms in our experiments, we train their offline component (e.g., the action sampler in BCQ [14] or the BC constraint in TD3-BC [13]) on the expanded MOCODA training distribution.*

## 5 Experiments

**Hypotheses**  Our experiments are aimed at finding support for two critical hypotheses:

- **H1** Dynamics models, especially ones sensitive to the local factorization, are able to generalize well in the MOCODA distribution.
- **H2** This out-of-distribution generalization can be leveraged via data augmentation to train an RL agent to solve out-of-distribution tasks.

Note that support for **H2** provides implicit support for **H1**.

**Domains**  We test MOCODA on two continuous control domains. First is a simple, but controlled, `2D Navigation` domain, where the agent must travel from one point in a square arena to another. States are 2D $(x, y)$ coordinates and actions are 2D $(\Delta x, \Delta y)$ vectors. In most of the state space, the sub-actions $\Delta x$ and $\Delta y$ affect only their respective coordinate. In the top right quadrant, however, the $\Delta x$ and $\Delta y$ sub-actions each affect *both* $x$ and $y$ coordinates, so that the environment is locally factored. The agent has access to empirical training data consisting of left-to-right and bottom-to-top trajectories that are restricted to a ⊐ shape of the state space (see the EMP distribution in Figure 4). We consider a target task where the agent must move from the bottom left to the top right. In this task there is sufficient empirical data to solve the task by following the ⊐ shape of the data, but learning the optimal policy of going directly via the diagonal requires out-of-distribution generalization.

Second, we test MOCODA in a challenging `HookSweep2` robotics domain based on Hook-Sweep [32], in which a Fetch robot must use a long hook to sweep two boxes to one side of the table (either toward or away from the agent). The boxes are initialized near the center of the table, and the empirical data contains trajectories of the agent sweeping exactly one box to one side of the table, leaving the other in the center. The target task requires the agent to generalize to states that it has never seen before (both boxes 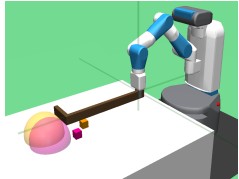 together on one side of the table). This is particularly challenging because the setup is entirely offline (no exploration), where poor out-of-distribution generalization typically requires special offline RL algorithms that constrain the agent's policy to the empirical distribution [34, 2, 31, 13].

**Directly comparing model generalization error.**  In the `2D Navigation` domain we have access to the ground truth dynamics, which allows us to directly compare generalization error on variety of

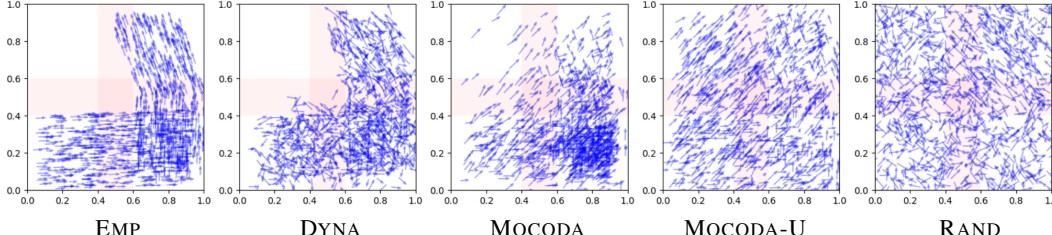

Figure 4: **2D Navigation Visualization.** (Best viewed with 2x zoom) Blue arrows represent transition samples as a vector from $(x_t, y_t)$ to $(x_{t+1}, y_{t+1})$. Shaded red areas mark the edges of the initial states of empirical trajectories and the center of the square. We see that 5-step rollouts (DYNA) do not fill in the center (needed for optimal policy), and fail to constrain actions to those that the model generalizes well on. For MOCODA, we see the effect of compounding dataset imbalance discussed in Subsection 4.1, which is resolved by MOCODA-U.

Table 1: **2D Navigation Dynamics Modeling Results:** Mean squared error $\pm$ std. dev. over 5 seeds, scaled by 1e2 for clarity (best model boldfaced). The locally factored model experienced less performance degradation out-of-distribution, and performed better on all distributions, except for the empirical distribution (EMP) itself.

| Model Architecture | Generalization Error (MSE $\times 1e2$) (lower is better) | | | | |
|---|---|---|---|---|---|
| | EMP | DYNA | RAND | **MoCoDA** | **MoCoDA-U** |
| Not Factored | **0.14 $\pm$ 0.04** | 2.41 $\pm$ 0.29 | 4.4 $\pm$ 0.31 | 0.95 $\pm$ 0.06 | 1.29 $\pm$ 0.15 |
| Globally Factored | 0.36 $\pm$ 0.01 | 2.09 $\pm$ 0.28 | 3.17 $\pm$ 0.3 | 0.41 $\pm$ 0.02 | 0.51 $\pm$ 0.02 |
| Locally Factored | 0.23 $\pm$ 0.1 | **1.47 $\pm$ 0.27** | **2.03 $\pm$ 0.19** | **0.33 $\pm$ 0.11** | **0.46 $\pm$ 0.11** |

Table 2: **2D Navigation Offline RL Results:** Average steps to completion $\pm$ std. dev. over 5 seeds for various RL algorithms (best distribution in each row boldfaced), where average steps was computed over the last 50 training epochs. Training on MOCODA and MOCODA-U improved performance in all cases. Interestingly, even using RAND improves performance, indicating the importance of training on out-of-distribution data. Note that this is an offline RL task, and so SAC (an algorithm designed for online RL) is not expected to perform well.

| RL Algorithm | Average Steps to Completion (lower is better) | | | | |
|---|---|---|---|---|---|
| | EMP | RAND | **MoCoDA** | **MoCoDA-U** | CoDA [51] |
| SAC (online RL) | 53.1 $\pm$ 9.8 | **27.6 $\pm$ 1.1** | 38.8 $\pm$ 18.3 | 41.3 $\pm$ 17.7 | 35.1 $\pm$ 18.1 |
| BCQ | 58.5 $\pm$ 10.1 | 31.7 $\pm$ 2.4 | **22.8 $\pm$ 0.4** | 24.8 $\pm$ 4.2 | 25.0 $\pm$ 0.4 |
| CQL | 45.8 $\pm$ 4.0 | 27.6 $\pm$ 1.3 | 22.8 $\pm$ 0.2 | **22.7 $\pm$ 0.3** | 23.6 $\pm$ 0.5 |
| TD3-BC | 40.0 $\pm$ 16.1 | 26.1 $\pm$ 0.8 | 21.0 $\pm$ 0.7 | **20.7 $\pm$ 0.8** | 21.4 $\pm$ 0.6 |

distributions, visualized in Figure 4. We compare three different model architectures: unfactored, globally factored (assuming that the $(x, \Delta x)$ and $(y, \Delta y)$ causal mechanisms are independent everywhere, which is not true in the top right quadrant), and locally factored. The models are each trained on a empirical dataset of 35000 transitions for up to 600 epochs, which is early stopped using a validation set of 5000 transitions. The results are shown in Table 1. We find strong support for **H1**: even given the simple dynamics of 2d Navigation, it is clear that the locally factored model is able to generalize better than a fully connected model, particularly on the MOCODA distribution, where performance degradation is minimal. We note that the DYNA distribution was formed by starting in EMP and doing 5-step rollouts with *random* actions. The random actions produce out-of-distribution data to which no model (not even the locally factored model) can generalize well to.

**Solving out-of-distribution tasks.** We apply the trained dynamics models to several base distributions and compare the performance of RL agents trained on each dataset. To ensure improvements are due to the augmented dataset and not agent architecture, we train several different algorithms, including: SAC [17], BCQ [14] (with DDPG [35]), CQL [31] and TD3-BC [13].

The results on 2D Navigation are shown in Table 2. We see that for all algorithms, the use of the MOCODA and MOCODA-U augmented datasets greatly improve the average step count, providing support for **H2** and suggesting that using these datasets allows the agents to learn to traverse the diagonal of the state space, even though it is out-of-distribution with respect to EMP. This is consistent with a qualitative assessment of the learned policies, which confirms that agents trained on the ⌐-shaped EMP distribution learn a ⌐-shaped policy, whereas agents trained on MOCODA and MOCODA-U learn the optimal (diagonal) policy.

The results on the more complex HookSweep2 environment, shown in Table 3, provide further support for **H2**. On this environment, only results for BCQ and TD3-BC are shown, as the other algorithms failed on all datasets. For HookSweep2 we used a prioritized MOCODA-P parent distribution, as follows: knowing that the target task involves placing two blocks, we applied rejection sampling to MOCODA to make the marginal distribution of the joint block positions approximately uniform over its support. The effect is to have good representation in all areas of the most important state features for the target task (the block positions). The visualization in Figure 5 makes clear why training on MOCODA or MOCODA-P was necessary in order to solve this task: the base EMP distribution simply does not have sufficient coverage of the goal space.

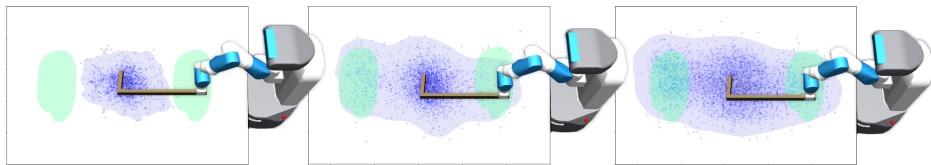

Figure 5: **HookSweep2 Visualization:** Stylized visualization of the distributions EMP (left), MOCODA (center), and MOCODA-P (right). Each figure can be understood as a top down view of the table, where a point is a plotted if the two blocks are close together on the table. The distribution EMP does not overlap with the green goal areas on the left and right, and so the agent is unable to learn. In the MOCODA distribution, the agent gets some success examples. In the MOCODA-P distribution, state-actions are reweighted so that the joint distribution of the two block positions is approximately uniform, leading to more evenly distributed coverage of the table.

Table 3: **HookSweep2 Offline RL Results:** Average success percentage ($\pm$ std. dev. over 3 seeds), where the average was computed over the last 50 training epochs. `SAC` and `CQL` (omitted) were unsuccessful with all datasets. We see that MOCODA was necessary for learning, and that results improve drastically with MOCODA-P, which re-balances MOCODA toward a uniform distribution in the box coordinates (see Figure 5). Additionally, we show results from an ablation, which generates the MoCoDA datasets using a fully connected dynamics model. While this still achieves some success, it demonstrates that using a locally-factored model is important for OOD generalization. In this case the more OOD MoCoDA-P distribution does not help, suggesting that the fully connected model is failing to produce useful OOD transitions.

| RLAlgorithm | **Average Success Rate** (higher is better) | | | | |
| --- | --- | --- | --- | --- | --- |
| | EMP | **MOCODA** | **MOCODA-P** | MOCODA (not factored) | MOCODA-P (not factored) |
| BCQ | $2.0 \pm 1.6$ | $20.7 \pm 4.1$ | $\mathbf{64.7 \pm 4.1}$ | $14.0 \pm 3.3$ | $15.3 \pm 4.1$ |
| TD3-BC | $0.7 \pm 0.9$ | $38.7 \pm 7.5$ | $\mathbf{84.0 \pm 2.8}$ | $29.3 \pm 3.8$ | $26.0 \pm 1.6$ |

## 6 Conclusion

In this paper, we tackled the challenging yet common setting where the available empirical data provides insufficient coverage of critical parts of the state space. Starting with the insight that locally factored transition models are capable of generalizing outside of the empirical distribution, we proposed MOCODA, a framework for augmenting available data using a controllable "parent distribution" and locally factored dynamics model. We find that adding augmented samples from MOCODA allows RL agents to learn policies that traverse states and actions never before seen in the experience buffer. Although our data augmentation is "model-based", the transition samples it produces are compatible with any downstream RL algorithm that consumes single-step transitions.

Future work might (1) explore methods for learning locally factorized representations, especially in environments with high-dimensional inputs (e.g., pixels) [22, 28], and consider how MOCODA might integrate with latent representations, (2) combine the insights presented here with learned predictors of out-of-distribution generalization (e.g., uncertainty-based prediction) [46], (3) create benchmark environments for entity-based RL [61] so that object-oriented methods and models can be better evaluated, and (4) explore different approaches to re-balancing the training distribution for learning on downstream tasks. With regards to direction (1), we note that asserting (or not) certain independence relationships may have fairness implications for datasets [47, 9] that should be kept in mind or explored. This is relevant also in regards to direction 4, as dataset re-balancing may result in (or fix) biases in the data [30]. Re-balancing schemes should be sensitive to this.

## Acknowledgments and Disclosure of Funding

We thank Jimmy Ba, Marc-Etienne Brunet, and Harris Chan for helpful comments and discussions. We also thank the anonymous reviewers for their feedback, which significantly improved the final manuscript. Silviu Pitis is supported by an NSERC CGS-D award. Animesh Garg is supported as a CIFAR AI chair, and by an NSERC Discovery Award, University of Toronto XSeed Grant and NSERC Exploration grant. Resources used in preparing this research were provided, in part, by the Province of Ontario, the Government of Canada, and companies sponsoring the Vector Institute.

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
