# A  Proof of Theorem 1

The dynamics model assumed is a maximum-likelihood, count-based model that has separate parameters for each causal mechanism, $P_{i,\theta}^{\mathcal{L}}$, in each local neighborhood. That is, for a given configuration of the parents $\text{Pa}_i = x$ in $P_{i,\theta}^{\mathcal{L}}$, we define count parameter $\theta_{ij}$ for the $j$-th possible child, $c_{ij}$, so that $P_{i,\theta}^{\mathcal{L}}(c_{ij} \mid x) = \theta_j / \sum_{k=1}^{|c_i|} \theta_k$.

We use the following two lemmas (see source material for proof):

**Lemma 1** (Proposition A.8 of Agarwal et al. [1]). *Let $z$ be a discrete random variable that takes values in $\{1, \ldots, d\}$, distributed according to $q$. We write $q$ as a vector where $\vec{q} = [Pr(z = j)]_{j=1}^d$. Assume we have $n$ i.i.d. samples, and that our empirical estimate of $\vec{q}$ is $[\vec{q}]_j = \sum_{i=1}^n \mathbf{1}[z_i = j]/n$. We have that $\forall \epsilon > 0$:*

$$Pr(\|\hat{q} - \vec{q}\|_2 \geq 1/\sqrt{n} + \epsilon) \leq e^{-n\epsilon^2}$$

*which implies that:*

$$Pr(\|\hat{q} - \vec{q}\|_1 \geq \sqrt{d}(1/\sqrt{n} + \epsilon)) \leq e^{-n\epsilon^2}$$

**Lemma 2** (Corollary 1 of Strehl [55]). *If for all states and actions, each model $P_{i,\theta}$ of $P_i$ is $\epsilon/k$ close to the ground truth in terms of the $\ell_1$ norm: $\|P_i(s,a) - P_{i,\theta}(s,a)\|_1 < \epsilon/k$, then the aggregate transition model $P_\theta$ is $\epsilon$ close to the ground truth transition model: $\|P(s,a) - P_\theta(s,a)\|_1 < \epsilon$.*

**Theorem 1.** *Let $n$ be the number of empirical samples used to train the model of each local causal mechanism $P_{i,\theta}^{\mathcal{L}}$ at each configuration of parents $\text{Pa}_i = x$. There exists positive constant $c$ such that, if*

$$n \geq \frac{ck^2|c_i|\log(|\mathcal{S}||\mathcal{A}|/\delta)}{\epsilon^2},$$

*then, with probability at least $1 - \delta$, we have:*

$$\max_{(s,a)} \|P(s,a) - P_\theta(s,a)\|_1 \leq \epsilon.$$

*Proof.* Applying Lemma 1, we have that for fixed parents $\text{Pa}_i = x$, wp. at least $1 - \delta$,

$$\|P_i(x) - P_{i,\theta}(x)\|_1 \leq c\sqrt{\frac{|c_i|\log(1/\delta)}{n}},$$

where $n$ is the number of independent samples used to train $P_{i,\theta}$ and $c$ is a positive constant. Now consider a fixed $(s,a)$, consisting of $k$ parent sets. Applying Lemma 2 we have that, wp. at least $1 - \delta$,

$$\|P(s,a) - P_\theta(s,a)\|_1 \leq ck\sqrt{\frac{|c_i|\log(1/\delta)}{n}}.$$

We apply the union bound across all states and actions to get that wp. at least $1 - \delta$,

$$\max_{(s,a)} \|P(s,a) - P_\theta(s,a)\|_1 \leq ck\sqrt{\frac{|c_i|\log(|S||A|/\delta)}{n}}.$$

The result follows by rearranging for $n$ and relabeling $c$. $\qquad \square$

To compare to full-state dynamics modeling, we can translate the sample complexity from the per-parent count $n$ to a total count $N$. Recall $m\Pi_i|c_i| = |\mathcal{S}|$, so that $|c_i| = (|\mathcal{S}|/m)^{1/k}$, and $m\Pi_i|\text{Pa}_i| \geq |\mathcal{S}||\mathcal{A}|$. We assume a small constant overlap factor $v \geq 1$, so that $|\text{Pa}_i| = v(|\mathcal{S}||\mathcal{A}|/m)^{1/k}$. We need the total number of component visits to be $n|\text{Pa}_i|km$, for a total of $nv(|\mathcal{S}||\mathcal{A}|/m)^{1/k}m$ state-action visits, assuming that parent set visits are allocated evenly, and noting that each state-action visit provides $k$ parent set visits. This gives:

**Corollary 1.** *To bound the error as above, we need to have*

$$N \geq \frac{cmk^2(|\mathcal{S}|^2|\mathcal{A}|/m^2)^{1/k}\log(|\mathcal{S}||\mathcal{A}|/\delta)}{\epsilon^2},$$

*total train samples, where we have absorbed the overlap factor $v$ into constant $c$.*

To extend this and adapt other results to our setting, we could now apply the Simulation Lemma [1] to bound the value difference given the model error, or alternatively, develop the theory in the direction of [55] and related work. However, we believe the core insights are already contained in Theorem 1 and Corollary 1.

# B    Implementation Details

Code is available at https://github.com/spitis/mocoda (using https://github.com/spitis/mrl for RL algorithms). There are numerous components involved that each have several different settings that were mostly just taken "as-is" or picked as reasonable defaults (e.g., using a layer size of 512 in most neural networks, or having 5 components in the MDN, or the specific implementation of rejection sampling for `Mocoda-U`). The best documentation for specific details is the code itself. As such, the implementation details below cover the broad strokes so that a reader might understand the general pipeline, and we refer the reader to the provided code for precise details.

## B.1    Causal Transition Structure and Parent Set Definitions

We implement the local causal model as a mask function $M$ that maps (state, action) tuples to an adjacency matrix of the causal structure. For example, in `2d Navigation`, the mask function was implemented as follows:

```
def Mask2dNavigation(input_tensor):
  """
  accepts B x num_sa_features, and returns B x num_parents x num_children
  """

  # base local mask
  mask = torch.tensor(
    [[1, 0],
     [0, 1],
     [1, 0],
     [0, 1]]).to(input_tensor.device)

  # change local mask in top right quadrant
  mask = mask[None].repeat((input_tensor.shape[0], 1, 1))
  mask[torch.logical_and(input_tensor[:,0] > 0.5, input_tensor[:, 1] > 0.5)] = 1

  return mask
```

As an example, the causal graph for the base local mask, which applies for most of the state space is shown in the figure 6. We used the base local graph to select the parent sets, in this case, $(x, \Delta x)$ and $(y, \Delta y)$.

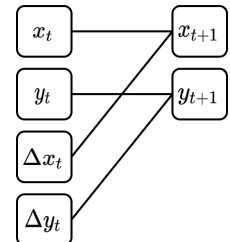

Figure 6: Causal graph for local mask

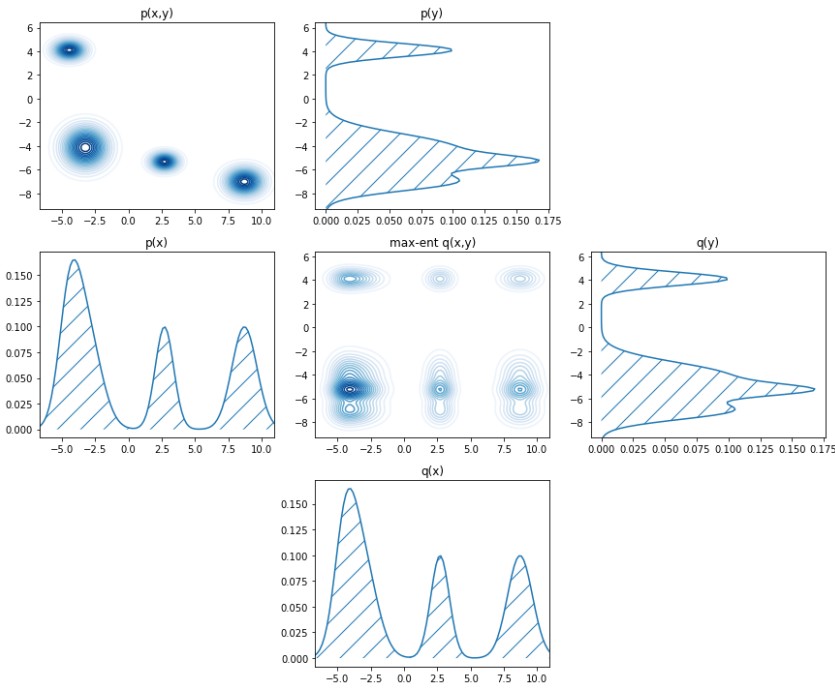

Figure 7: Hypothetical 2D illustration of the GMM-based parent set sampler. It is assumed that there are two non-overlapping parent sets $\{x\}$ and $\{y\}$, but that $x$ and $y$ exhibit dependence in the empirical data. We fit a GMM to each marginal $P(x)$ and $P(y)$ and sample from them independently to get $Q(x, y)$, which has the same marginal distributions (so that the components in a locally factored dynamics model will generalize), but eliminates the spurious dependence in the empirical data.

## B.2 Augmented Parent Distribution

To sample from the augmented parent distribution in Step 1 of MoCoDA, we use the Gaussian Mixture Model (GMM) based approach described in the main text. The advantage of this approach is that we can easily do conditional sampling in case of overlapping parent sets. For a given local subset $\mathcal{L}$, we fit a separate GMM to the marginal of each parent set, as it appears in the empirical distribution for $\mathcal{L}$. To generate a new sample, we optionally shuffle the GMMs, and then sample from one GMM at a time, conditioning on any already generated features. This process eliminates any spurious correlations between features that are not part of the same parent set, and thus results in the maximum-entropy, marginal matching distribution.

In cases of multiple local neighborhoods, $\mathcal{L}_1, \mathcal{L}_2, \ldots$, one should respect the boundaries of the current local subset $\mathcal{L}$ during both training *and* generation. If a sample generated with the GMM for $\mathcal{L}$ falls outside of $\mathcal{L}$, that sample should be rejected, as the local causal structure is no longer valid, and the generalization guarantee for the locally factored model no longer holds.

As the local factorization in our experiments is quite simple, we did not stratify the GMM generator, and instead used a single GMM generator for the sparsest local causal structure. In the case of 2d Navigation this did not generate any data that was out-of-distribution for the locally factored model components (as the agent's policy was consistent in all local neighborhoods). In the case of HookSweep2, there was a bit of locally out-of-distribution data in the local subspace in which there is a block collision; however, most of this data is unreachable as it involves overlapping blocks, and we obtained strong results even with this shortcut.

## B.3 Dynamics Models

Our experiments used three different dynamics models. In each case, we used an ensemble of 5 base models, described below. The base models output a Gaussian mean and variance for each output variable and are trained independently via a negative log likelihood loss. All models are trained using Adam Optimizer [26].

A. **Unfactored:** The base model is a fully connected neural network with ReLU activations (MLP).

B. **Globally Factored:** The base model has one MLP for each causal mechanism in the sparsest local graph. For both `2d Navigation` and `HookSweep2` the sparest local graph has two components, so the base global model is composed of two MLPs.

C. **Locally factored:** The base model is designed as follows. For each child node, $c_i$, there is a separately parameterized single MLP that is preceded by a "Masked Composer" module. The Masked Composer applies a single layer MLP (linear transform followed by ReLU) to each root node, $r_i$ (each parent set has several nodes), to obtain embeddings $\varepsilon_i(r_i)$. The $i$-th column of the mask is used to zero out the corresponding embeddings which are then summed, $\sum_i M_{ij}\varepsilon_i(r_i)$, and the result is passed as an input to the MLP.

This architecture works (and enforces local factorization), but is likely poor, because it does not take advantage of potentially useful shared representations between parent nodes across children (since there is a separately parameterized Masked Composer for each child). A better architecture would likely use a single parameterization for a single, possibly deeper Masked Composer. As this is not the focus of our contribution, we stuck with simple model, as it "just worked" for purposes of our experiments.

### B.4 Training Data for the RL Algorithm

This varied by experiment, and is described in the next Section. Notably, we divided the standard deviation returned by our dynamics models by a factor of three when generating data to avoid data that was too far out of distribution.

### B.5 Reinforcement Learning Algorithms

We use Modular RL [50], adding three offline RL [34] algorithms: BCQ [14], CQL [31] & TD3-BC [13].

The BCQ implementation uses DDPG [35]. For the generative model we use a Mixture Density Network (MDN) [6] with 5 components, that produces 20 action samples at each call (both during test rollouts and when creating critic targets). The MDN was trained for 1000 batches with batch size of 2000. We did not use a perturbation model.

The CQL implementation uses SAC [17]. Rewards in our environments are sparse, and so value targets can be accurately clipped between two values (depends on the discount factor). CQL balances two losses: a penalty for Q-values of some non-behavioral distribution/policy (we use a random policy), and a bonus for the Q-values behavioral actions. We use an L1 penalty toward the lower end of the value target clipping range, and an L1 bonus toward the higher end of the value target clipping range. We then multiply that by a minimum Q coefficient, as in the original CQL implementation.

The TD3-BC implementation follows Fujimoto and Gu [13].

## C   Experimental Details

### C.1   2D Navigation

In this environment, the agent must travel from one point in a square arena to another. States are 2D $(x, y)$ coordinates and actions are 2D $(\Delta x, \Delta y)$ vectors.

```
observation_space = spaces.Box(np.zeros((2,)), np.ones((2,)), dtype=np.float32)
action_space = spaces.Box(-np.ones((2,)), np.ones((2,)), dtype=np.float32)
```

Episodes run for up to 70 steps. Rewards are sparse, with a -1 reward everywhere except the goal, where reward is 0. In most of the state space, the sub-actions $\Delta x$ and $\Delta y$ affect only their respective coordinate. In the top right quadrant, however, the $\Delta x$ and $\Delta y$ sub-actions each affect *both* $x$ and $y$ coordinates, so that the environment is locally factored. The two causal graphs are as follows:

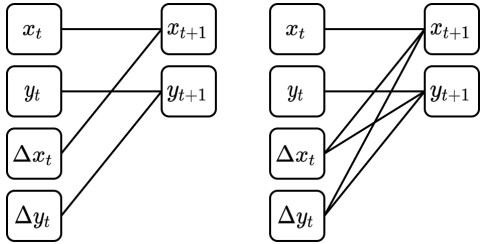

The graph on the right applies only in the top-right quadrant; otherwise the graph on the left applies. The graph on the left has non-overlapping parent sets $(x, \Delta x)$ and $(y, \Delta y)$. The graph on the right has overlapping parent sets $(x, \Delta x, \Delta y)$ and $(y, \Delta x, \Delta y)$.

The agent has access to an empirical dataset consisting of left-to-right & bottom-to-top trajectories (20,000 transitions of each type):

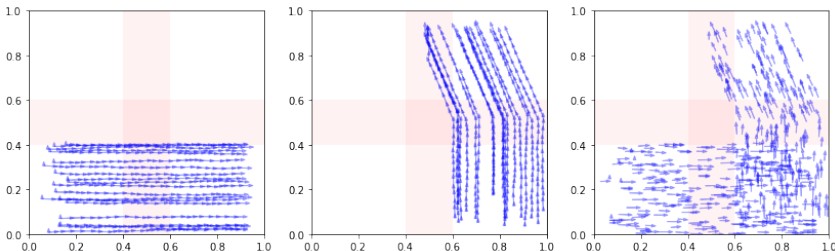

Figure 8: **Left/Middle:** Random samples of the two types of trajectories the agent has access to. **Right:** Random sample of transitions from this empirical dataset.

We consider a target task where the agent must move from the bottom left to the top right. In this task there is sufficient empirical data to solve the task by following the ⌐ shape of the data, but learning the optimal policy of going directly via the diagonal requires out-of-distribution generalization.

For 2d Navigation, we generated the MOCODA distribution by fitting a GMM generator as described in the previous Section. Each GMM (one for each parent set) had 32 components, and was fit using expectation maximization. To obtain MOCODA-U, we implemented rejection sampling by using a KDE density estimator is as follows:

```
def prune_to_uniform(proposals, target_size=12000.):
  from sklearn.neighbors import KernelDensity
  sample = proposals[-10000:]

  fmap = lambda s: s[:, :2]
  K = KernelDensity(bandwidth=0.05)
  K.fit(fmap(sample))
  scores = K.score_samples(fmap(proposals))
  scores = np.maximum(scores, np.log(0.01))
  scores = (1. / np.exp(scores))
  scores = scores / scores.mean()  * (target_size / len(proposals))

  return proposals[np.random.uniform(size=scores.shape) < scores]
```

The dynamics models each had 2 layers of 256 neurons and were trained with a batch size of 512 and learning rate of 1e-4. Hyperparameters were not tuned once a working setting was found. Of the 40K empirical samples, 35K were used for training, and 5000 for validation. The models were trained for 600 epochs, with early stopping used in the last 50 epochs to find a locally optimal stopping point.

Augmented datasets of 200K samples were generated. In each case except EMP, 40K were the original empirical dataset (thus 160K new samples were generated by applying the dynamics model to samples from the augmented distribution). In case of EMP, the 40K original samples were simply repeated 5 times to get a size 200K dataset. The locally factored network was used to generate the augmented datasets.

These augmented distributions were then used to train the downstream RL agents. The agent algorithms used a discount factor of 0.98, a target cutoff range of (-50, 0), batch size of 500, and used 2 layers of 512 neurons in both actor and critic networks. The agents were trained for 25K batches (for a total of 62.5 passes over the dataset).

For 2D `Navigation` we ran 5 seeds, which all yielded similar results. For each seed we trained new parent set samplers and generated new augmented datasets.

## C.2 HookSweep2

`HookSweep2` is a challenging robotics domain based on Hook-Sweep [32], in which a Fetch robot must use a long hook to sweep two boxes to one side of the table (either toward or away from the agent). States, excluding the goal, are 16 dimensional continuous vectors. Goals are 6 dimensions. The agents all concatenate the goal to the state, and so operate on 22 dimensional states. The action space is a 4 dimensional continuous vector.

The environment contains two boxes that are initialized near the center of the table.

The empirical data contains 1M transitions from trajectories of an expert agent sweeping exactly one box to one side of the table, leaving the other in the center. The target task requires the agent to sweep *both* boxes together to one side of the table. This is particularly challenging because the setup is entirely offline (no exploration), where poor out-of-distribution generalization typically requires special offline RL algorithms that constrain the agent's policy to the empirical distribution [34, 2, 31, 13].

Episodes run for 75 steps. Rewards are dense, but structured similarly to a sparse reward, with a base reward of -1 everywhere except the goal and a reward of 0 at the goal. Additional small rewards are given if the agent keeps the hook near the table (this was required to obtain natural movements from the trained expert agent).

In this environment, *we did not have the ground truth causal graph*, and so a heuristic was used. The heuristic (wrongly) assumes that the agent/hook *always* causes each of the next object position (hook and objects are always entangled), even though this is only true when the hook and the objects are touching. The heuristic considers the two boxes to be separate whenever they are further than 5cm from each other. Here is the implementation of the heuristic:

```
def MaskHookSweep2(input_tensor):

    # base local mask for when boxes are far apart
    mask = torch.tensor(
      [[1, 1, 1],
       [1, 1, 0],
       [1, 0, 1],
       [1, 1, 1]]
       ).to(input_tensor.device)
    mask = mask[None].repeat((input_tensor.shape[0], 1, 1))

    # change local mask when boxes are close to each other
    mask[torch.sum(torch.abs(input_tensor[:,O1X:O1X+2] -\
        input_tensor[:,O2X:O2X+2]), axis=1) < 0.05] = 1

    return mask
```

where the state-action components are (gripper, box1, box2, action). This heuristic returns the following two causal graphs (note that goals are not part of the dynamics, and are separately labeled using random goal samples from the environment):

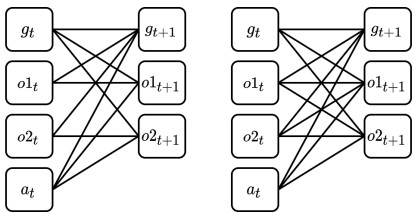

The parent sets are (g, o1, a) and (g, o2, a) for the first graph, and (g, o1, o2, a) in the second graph.

For `HookSweep2`, the generation of the MOCODA distribution is identical to how it was generated in `2d Navigation` (see previous subsection). To obtain MOCODA-P, we implemented rejection sampling as follows:

```python
def prune_to_uniform2(proposals, target_size=12000., smaller=True):
  proposals = proposals[np.linalg.norm(proposals[:,O1X:O1X+2] - proposals[:,O2X:O2X+2], axis=-1) < 0.3]
  sample = proposals[-5000:]

  fmap = lambda s: s[:,[O1X,O1X+1,O2X,O2X+1]]
  K = KernelDensity(bandwidth=0.05)
  K.fit(fmap(sample))
  scores = K.score_samples(fmap(proposals))
  scores = np.maximum(scores, np.log(0.05))
  scores = (1. / np.exp(scores))
  if np.minimum(scores, 1).sum() > 10000:
  while np.minimum(scores, 1).sum() > 10000:
    scores = scores * 0.99
  else:
  while np.minimum(scores, 1).sum() < 10000:
    scores = scores / 0.99

  return proposals[np.random.uniform(size=scores.shape) < scores]
```

The key difference to the `2d Navigation` is the definition of the `fmap` function, which defines the feature map under which the density is computed for rejection sampling.

The dynamics models for `HookSweep2` each had 2 layers of 512 neurons and were trained with a batch size of 512 and learning rate of 2e-4. Hyperparamters were not tuned once a working setting was found (learning rate was increased to make training slightly faster). Of 1M empirical samples, 5000 were used for validation. The models were trained for 4000 epochs, where each epoch involved 40K random samples, with early stopping used in the last 50 epochs to find a locally optimal stopping point.

Augmented datasets of 5M samples were generated. In each case except EMP, 1M were the original empirical dataset (thus 4M new samples were generated). In the case of EMP, the 1M original samplers were simply repeated 5 times to get the full augmented dataset. The locally factored network was used to generate the augmented datasets.

These augmented distributions were then used to train the downstream RL agents. The agent algorithms were the same as for `2d Navigation`, except that they used 3 layers of 512 neurons in both actor and critic networks. The agents were trained for 1M steps with batch size 500 (for a total of 100 passes over the dataset).

### C.3 Licenses and Compute

All experiments were run on a modern desktop CPU and a NVIDIA GTX 1080 Ti GPU.

Code and assets are available under Apache and MIT licenses from Mujoco, OpenAI Gym, AC-Teach [32], and Modular RL [50] repositories. The implementations used in this paper will be released upon acceptance under an open source license.

## D   Further Discussion of Broader Impacts

MOCODA uses causally-motivated data augmentation to tackle sequential decision making problems where the available experience data may not be sufficient to find an optimal policy for the task at hand. While we have thusfar applied this approach to continuous control problems, there are a large body of problems that share this general motivation, where long-term fairness and robustness may be a central concern [10]. In these cases, the causal assumptions used to implement MOCODA deserve extra care and external scrutiny. For such problems, the structure of the state space may include sensitive and/or socially-ascribed attributes of groups and individuals (which cannot be directly intervened upon), so any graphical causal will involve normative assumptions about the environment in which the agent is embedded [18].

## E   MOCODA sampling pseudocode

Algorithm 1 shows the pseudocode for MOCODA sampling for a FMDP, where the causal structure of the transition dynamics is assumed known. For simplicity of exposition we describe the case where

**Algorithm 1** MOCODA for FMDPs, assuming hand-specified dynamics factorization

1: **function** GENERATEMOCODADATA(N):
2:   **Input:** observed transition dataset $(s, a, s') \in \mathcal{D}$
3:   **Input:** causal structure of transition dynamics $\mathcal{G} := \{(i, \mathrm{Pa}(i)) \ \forall \ i \in N_s\}$ a.k.a. "parent sets"
4:       NOTE: $\mathrm{Pa}(i)$ is shorthand for $\{j : s_j \in \mathrm{CausalParent}(s_i')\}$
5:       NOTE: $\mathrm{Pa}(i) \subset [N_s + N_a]$ can index into states or actions

6:   $\mathcal{D}_{tr}, \mathcal{D}_{va} = \texttt{train\_val\_split}(\mathcal{D})$                                                  ▷ Split data
7:   $\theta := \text{TRAINGMMPARENTSMODEL}(\mathcal{D}_{tr}, \mathcal{D}_{va}, \mathcal{G})$        ▷ train GMM parent distribution $\tilde{P}_\theta(s, a)$
8:   $\phi := \text{TRAINFACTOREDDYNAMICS}(\mathcal{D}_{tr}, \mathcal{D}_{va}, \mathcal{G})$        ▷ train factored dynamics model $P_\phi(s'|s, a)$
9:   **return** SAMPLEAUGMENTEDDATASET(N, $\theta$, $\phi$, $\mathcal{G}$)

10: **function** TRAINGMMPARENTSMODEL($\mathcal{D}_{tr}, \mathcal{D}_{va}, \mathcal{G}$):
11:   **for** $(i, \mathrm{Pa}(i)) \in \mathcal{G} :$ **do**                                       ▷ iterate over parent sets for each child
12:       $\{\mu_{\mathrm{Pa}(i)}^k, \Sigma_{\mathrm{Pa}(i)}^k, \gamma_{\mathrm{Pa}(i)}^k\} = \texttt{init\_gmm\_params}(N_k)$        ▷ NOTE: *only* for this parent set
13:       $\mathcal{D}_{\mathrm{Pa}(i)}^{tr}(s, a) := \{(s[\mathrm{Pa}(i)], a[\mathrm{Pa}(i)\%N_s]) \forall (s, a, s') \in \mathcal{D}^{tr}\}$
14:       $\mathcal{D}_{\mathrm{Pa}(i)}^{va}(s, a) := \{(s[\mathrm{Pa}(i)], a[\mathrm{Pa}(i)\%N_s]) \forall (s, a, s') \in \mathcal{D}^{va}\}$        ▷ subsample relevant dims
15:       **while** $\texttt{not\_converged}(\mathrm{GMM}(\cdot; \mu_{\mathrm{Pa}(i)}^k, \Sigma_{\mathrm{Pa}(i)}^k, \gamma_{\mathrm{Pa}(i)}^k), \mathcal{D}_{\mathrm{Pa}(i)}^{va}(s, a))$ **do**
16:           $\{\mu_{\mathrm{Pa}(i)}^k, \Sigma_{\mathrm{Pa}(i)}^k, \gamma_{\mathrm{Pa}(i)}^k\} \leftarrow \texttt{update\_gmm\_params}(\{\mu_{\mathrm{Pa}(i)}^k, \Sigma_{\mathrm{Pa}(i)}^k, \gamma_{\mathrm{Pa}(i)}^k\}, \mathcal{D}_{\mathrm{Pa}(i)}^{tr}(s, a))$
17:       $\theta.\texttt{append}(\{\mu_{\mathrm{Pa}(i)}^k, \Sigma_{\mathrm{Pa}(i)}^k, \gamma_{\mathrm{Pa}(i)}^k\})$
18:   **return** $\theta$

19: **function** TRAINFACTOREDDYNAMICS($\mathcal{D}_{tr}, \mathcal{D}_{va}, \mathcal{G}$):
20:   **for** $(i, \mathrm{Pa}(i)) \in \mathcal{G} :$ **do**                                       ▷ iterate over parent sets for each child
21:       $\phi_i = \texttt{init\_mlp\_params}()$                                       ▷ NOTE: *only* for this parent set
22:       $\mathcal{D}_{\mathrm{Pa}(i)}^{tr}(s, a, s') := \{(s[\mathrm{Pa}(i)], a[\mathrm{Pa}(i)\%N_s], s'[i]) \forall (s, a, s') \in \mathcal{D}^{tr}\}$
23:       $\mathcal{D}_{\mathrm{Pa}(i)}^{va}(s, a, s') := \{(s[\mathrm{Pa}(i)], a[\mathrm{Pa}(i)\%N_s], s'[i]) \forall (s, a, s') \in \mathcal{D}^{va}\}$        ▷ subsample relevant dims
24:       **while** $\texttt{not\_converged}(\mathrm{MLP}(\cdot|\cdot; \phi_i), \mathcal{D}_{\mathrm{Pa}(i)}^{va}(s, a, s'))$ **do**
25:           $\phi_i \leftarrow \texttt{update\_mlp\_params}(\phi_i, \mathcal{D}_{\mathrm{Pa}(i)}^{tr}(s, a, s'))$
26:   **return** $\phi$

27: **function** SAMPLEAUGMENTEDDATA(N, $\theta$, $\phi$, $\mathcal{G}$)
28:   **for** $\_ \in \texttt{range}(N):$ **do**
   ▷ sample parent data, i.e. $(\tilde{s}, \tilde{a})$: sequentially sample the parent set GMMs, conditioning
   ▷ each GMM on previous samples to handle any overlap between parent sets
29:       $\mathcal{AS} := \{\ \}$                                       ▷ define an "already sampled" set to track any parent set overlap
30:       $\tilde{s} := [\ ]; \tilde{a} := [\ ]$
31:       **for** $i \in \texttt{range}(N_s):$ **do**
32:           **if** $\mathrm{Pa}(i) \cap \mathcal{AS} = \emptyset :$ **then**                                       ▷ no vars in this parent set already sampled
33:               $(\tilde{s}_{\mathrm{Pa}(i)}, \tilde{a}_{\mathrm{Pa}(i)}) \sim \mathrm{GMM}(\cdot; \mu_{\mathrm{Pa}(i)}^k, \Sigma_{\mathrm{Pa}(i)}^k, \gamma_{\mathrm{Pa}(i)}^k)$
34:               $\tilde{s}.\texttt{extend}(\tilde{s}_{\mathrm{Pa}(i)})$
35:               $\tilde{a}.\texttt{extend}(\tilde{a}_{\mathrm{Pa}(i)})$
36:           **else**                       ▷ some vars in this parent set already sampled and must be conditioned on
   ▷ condition this GMM already-sampled vars, then sample remaining vars
37:               $(\tilde{s}_{\mathrm{Pa}(i)\setminus\mathcal{AS}}, \tilde{a}_{\mathrm{Pa}(i)\setminus\mathcal{AS}}) \sim \mathrm{GMM}(\cdot|\mathcal{AS}; \mu_{\mathrm{Pa}(i)}^k, \Sigma_{\mathrm{Pa}(i)}^k, \gamma_{\mathrm{Pa}(i)}^k)$
   ▷ NOTE: this sampling is easily realized by conditioning each Gaussian component
   ▷ and updating mixture components in proportion to density of already-sampled vars
38:               $\tilde{s}.\texttt{extend}(\tilde{s}_{\mathrm{Pa}(i)\setminus\mathcal{AS}})$
39:               $\tilde{a}.\texttt{extend}(\tilde{a}_{\mathrm{Pa}(i)\setminus\mathcal{AS}})$
40:           $\mathcal{AS} \leftarrow \mathcal{AS} \cup \mathrm{Pa}(i)$
   ▷ sample next states, i.e. $\tilde{s}'|(\tilde{s}, \tilde{a})$: sequentially sample each "factor" in the factorized dynamics
41:       $\tilde{s}' := [\ ]$
42:       **for** $i \in \texttt{range}(N_s):$ **do**
43:           $\tilde{s}_i' \sim \mathrm{MLP}(\cdot|\tilde{s}, \tilde{a}; \phi_i)$
44:           $\tilde{s}'.\texttt{append}(\tilde{s}_i')$
   ▷ assemble transition
45:       $\tilde{s} = \texttt{array}(\tilde{s}); \tilde{a} = \texttt{array}(\tilde{a}); \tilde{s}' = \texttt{array}(\tilde{s}')$
46:       $\tilde{\mathcal{D}}.\texttt{append}((\tilde{s}, \tilde{a}, \tilde{s}'))$
47:   **return** $\tilde{\mathcal{D}}$

each "factor" in the factorized dynamics is modeled using an MLP, which corresponds to the "Globally Factored" model architecture referred to in Table 1. Realizing the "Locally Factored" architecture is simply a matter of replacing $\text{MLP}(\cdot)$ in the pseudocode with $(\text{MLP} \circ \text{MaskedComposer})(\cdot)$ described in Section B.3.

Implementing MOCODA sampling for a Local Causal Model rather than an FMDP is also a straightforward extension. The dynamics modeling is the same (but for the dynamics being conditioned on $\mathcal{L}$). The parent sampling procedure is described in B.2.

## F    Relation to Causal Inference and Counterfactual Reasoning

Although MOCODA leverages a causal structure on the transition dynamics, and although it is possible that the models used by MOCODA could be used for a limited form of causal inference (described below), the actual MOCODA algorithm does not do Pearl-style "counterfactual reasoning" [49] when sampling new transitions. This is because the generic algorithm presented in Algorithm 1 uses the (augmented) parent model $\tilde{P}_\theta(s, a)$ and dynamics model $P_\phi(s'|s, a)$ together to sample entire transitions *de novo*, whereas Pearl-style counterfactuals ask "what if $X'$ happened instead of $X$ (given $Y$ was observed)?". Answering this latter question in the SCM framework involves inference over exogenous noise variables, and typically results in a partial relabeling of the data with the counterfactual result. In MOCODA, there are no explicit noise variables, and the noise is implicit in the parent and dynamics models.

Nevertheless, one could do a form of Pearl-style counterfactual reasoning using the MOCODA models by considering counterfactual "what if" questions for a subset of all parent variables, rather than all variables simultaneously. In this case, the parent model could be used to conditionally resample any unspecified parent variables (conditioning the remaining variables on factual observations), and the dynamics model could be used to resample any affected causal mechanisms (keeping the rest of the transition, and therefore any noise implicit in the rest of the transition, fixed).

When carrying out causal inference or counterfactual reasoning, a central concern is *identifiability*: under what assumptions are inferences or counterfactual samples produced by an algorithm said to be unique? Demonstrating identifiability, say of counterfactual transitions, typically requires the introduction of further assumptions over the structural functions themselves, which can be realized in practice through the use of specialized network architectures (which we have not employed in our experiments). For example, Oberst and Sontag [44] extended the monotonicity assumption developed for binary outcomes [48] to categorical transition dynamics, enabling identifiability analyses to discrete MDPs. Lu et al. [38] applied SCM dynamics to data augmentation in continuous sample spaces, and discussed the conditions under which the generated transitions are uniquely identifiable counterfactual samples. Without appealing to such approaches, we note that the discrete setting used for analysis in Section 3 could be further constrained by assuming deterministic and invertible dynamics, which would yield identifiable counterfactual sampling using the MOCODA models. However, these restrictive assumptions preclude most practical settings.