# OpenReview forum: "MoCoDA: Model-based Counterfactual Data Augmentation"
_NeurIPS.cc/2022/Conference — NeurIPS 2022 Accept_

### Official Review · Reviewer_8Yp5 · 2022-07-10

**Rating:** 7
**Confidence:** 4
**Soundness:** 3 good
**Presentation:** 3 good
**Contribution:** 3 good

**Summary:**

This paper proposes a model-based data augmentation approach named MoCoDa. The approach exploits the local factorization in state-action space, inspired by causality and factored MDP literature, to generate transitions for policy learning. The authors provided the theoretical analysis of sample complexity on the local factored model, and meanwhile, the empirical results on two continuous control benchmarks verify the hypotheses made by the main paper.

**Questions:**

**Please note that running any of the mentioned ideas or datasets is not a must during the rebuttal phase. Some discussion is highly appreciated.**

#### **1. About the entangled long horizon structure**

The authors said the long horizon prediction could no longer leverage the sparsity of causal graphs. I can roughly understand what the authors meant here. However, suppose we can learn a good disentangled representation of "decision-making-relevant" and "decision-making-irrelevant" state decomposing in the graph. In that case, we can leverage the sparsity of the causal graphs. Similar ideas have been proposed in [1], where they also learn the connections between states and rewards using masks. And those variables (states) having direct connections to future rewards are decision-making-relevant. Then we can leverage these states for down-streaming policy learning.

#### **2. Graphs in long horizon structure**

The authors said that temporal entanglement might happen. Why not consider using the non-stationary graphs with temporal changes to describe the interaction skeletons?

#### **3. More complicated evaluation senarios**

Can the proposed method be applied to other complicated OOD generalization tasks, where the distribution shift is more natural and practical? e.g., Procgen Benchmark [2], different Atari games, etc.


#### **References**

[1] Huang, Biwei, et al. "Action-sufficient state representation learning for control with structural constraints." International conference on machine learning, 2022.

[2] Cobbe, Karl, et al. "Leveraging procedural generation to benchmark reinforcement learning." International conference on machine learning. PMLR, 2020.


**Limitations:**

Most of the limitations are mentioned in the weakness parts. I think the authors can improve the work by giving more clear clarifications on causality contexts. I think the paper is a good submission, and I will increase my score if the authors provide supporting analysis on the above weaknesses.

**Strengths And Weaknesses:**

## *Strengths*

#### **1. Technical soundness and significance**

In general, the paper delivers a novel approach, inspired by causality and factored MDP, to effectively generate transition samples for RL. The method is technically sound and would potentially benefit the fields of RL, causal representation learning, etc.

#### **2. Empirical verification**

The paper includes two evaluation tasks on continuous control domains. The results indicate that the model can learn the underlying dynamic process and improve the OOD generalization on novel tasks. The quantitative results (Table 1-3) and visualization (Fig. 4-5) can verify the effectiveness.

#### **3. Presentation**

The overall presentation is clear and easy to follow, although it can be further improved by addressing a few minor points (mentioned below in the weakness sections). The appendix supports the main paper and provides most of the required details.

## *Weaknesses*

#### **1. About causality**

##### *1.1. The concept of "counterfactual" is vague in the paper*

Usually, in the context of causality, counterfactual reasoning means you must reconstruct the exogenous noise factors in the data generating process (see Pearl's Causal ladder in [1-2]). While in this paper, some critical and crucial points are missing. It would be highly appreciated if the authors could make some discussion and analysis (if any) in the rebuttal and final version of this paper. The authors can refer to one of the existing works on data augmentation using counterfactual learning for RL [3].

##### *1.2. Identifiability of the (learned or given) causal graphs*

Similarly, we usually need to guarantee that the graph is identifiable in the causality literature. It would be better to give proof or analysis of identifiability since the authors assume the given or learned graphs are essentially causal graphs.

#### **2. Missing comparisons with similar approaches**

Some other approaches [4-10] also aim to learn the factored masks for RL to improve the sample efficiency or few/zero-shot generalization. It would be better for the authors to discuss them in the updated version. **Please note that running them in the rebuttal phase is not a must. The discussion and analysis would be highly appreciated in the rebuttal.**

#### **3. Missing references**

**Disclaimer: All mentioned works are critical and representative works in different related areas, and none of them is done by me.**

##### *3.1 Factored MDP for deep RL: [4-10]*

#### *3.2 Causal representation learning: [11]*

#### *3.3 Sparse/independent  (causal) mechanisms: [12-13]*

#### *3.4 Causal discovery in dynamic systems [14-15]*


#### **4. Presentations**

##### 4.1 *About CoDA*

Since the work is built upon CoDA, it is better to have a short background or preliminary section on CoDA (about motivation, model, and important remarks/conclusions) and the relevance and differences with MoCoDA. I noticed that the authors had provided analysis in Section 4, but giving them in the early sections will help authors to understand these two works better.

##### 4.2 *About masked dynamic models*

The current content on deriving the masks is vague. The authors can consider moving some contents in Appendix B.2 (well-written and easy to understand) into the main paper for better readability.

##### 4.3 **Adding more related works sections**

Adding more related works sections on causality (structural causal model) and causal RL, data augmentation for RL, and OOD generalization for RL would be helpful.

##### 4.4 **Minor typos**

a. Line 43: they too learn to generalize ... [check grammar]

b. Line 237: missing the right bracket






#### **References**

[1]. Pearl, Judea. Causality. Cambridge university press, 2009.

[2]. Powell, Stephen. "The Book of Why: The New Science of Cause and Effect. Pearl, Judea, and Dana Mackenzie. 2018. Hachette UK." Journal of MultiDisciplinary Evaluation 14.31 (2008): 47-54.

[3] Lu, Chaochao, et al. "Sample-efficient reinforcement learning via counterfactual-based data augmentation." NeurIPS workshop on Offline RL (2020).

[4] Hallak, Assaf, et al. "Off-policy model-based learning under unknown factored dynamics." International Conference on Machine Learning. PMLR, 2015.

[5] Balaji, Bharathan et al. Factoredrl: Leveraging factored graphs for deep reinforcement learning. NeurIPS 2020 Deep Reinforcement Learning Workshop, 2020.

[6] Loynd, Ricky, et al. "Working memory graphs." International conference on machine learning. PMLR, 2020.

[7] Wang, Tingwu, et al. "Nervenet: Learning structured policy with graph neural networks." International conference on learning representations. 2018.

[8] Artem Zholus, Yaroslav Ivchenkov, and Aleksandr Panov. Factorized world models for learning causal relationships. In ICLR2022 Workshop on the Elements of Reasoning: Objects, Structure and Causality, 2022.

[9] Shagun Sodhani, Sergey Levine, and Amy Zhang. Improving generalization with approximate factored value functions. In ICLR2022 Workshop on the Elements of Reasoning: Objects, Structure and Causality, 2022.

[10] Huang, Biwei, et al. "Action-sufficient state representation learning for control with structural constraints." International conference on machine learning, 2022.

[11] Schölkopf, Bernhard, et al. "Toward causal representation learning." Proceedings of the IEEE 109.5 (2021): 612-634.

[12] Goyal, Anirudh, et al. "Recurrent independent mechanisms." ICLR 2021.

[13] Madan, Kanika, et al. "Fast and slow learning of recurrent independent mechanisms."  ICLR 2021.

[14] Bongers, Stephan, Tineke Blom, and Joris Mooij. "Causal modeling of dynamical systems." arXiv preprint arXiv:1803.08784 (2018).

[15] Yao, Weiran, Guangyi Chen, and Kun Zhang. "Learning Latent Causal Dynamics." arXiv preprint arXiv:2202.04828 (2022).

---

> ### Author Response · Authors · 2022-08-01
> **Response to Reviewer 8Yp5 (1/2)**
>
> Thank you for your detailed and considered review, and for pointing us to relevant prior work on related topics. We address your comments in several parts:
>
> **1. About Causality**
>
> We agree that the paper will be improved by a further discussion of causal inference and counterfactual reasoning, specifically around the issue of identifiability, and we appreciate the pointers you provide to relevant papers on the topic. We have included such a discussion in the revision [Appendix F]. We clarify that while MoCoDA leverages a causal structure on the transition dynamics, and although it is possible that the models used by MoCoDA could be used for a limited form of causal inference (described in Appendix F), the actual MoCoDA algorithm does not do Pearl-style ``counterfactual reasoning'' [1] when sampling new transitions.
>
> It is possible to adapt the MoCoDA models to realize a formal counterfactual sampling procedure. In the setting of tabular RL over discrete states and actions, identifiable counterfactuals are trivially produced by assuming deterministic and invertible dynamics. One thing to note is that the ‘intervention’ in the second step of  abduction-action-prediction step is over the full inputs to a single causal mechanism (i.e. parent set), rather than a specific action as in Lu et al 2020, and are thus less directly interpretable as an answer to the question ‘what would happen if the agent had taken a different action?’. In any case, use of deterministic and invertible dynamics sidesteps the need to model exogenous noise explicitly, while the counts-based generative models are valid for any choice of (s,a) sampled by the parent model P_\theta(s,a), because the model for each parent set is ‘trained’ (i.e., counted) over its proper support with sufficient number of samples [lines 151–153]. We admit these assumptions preclude most practical settings. Fortunately we could make use of previous works in this area to extend the analysis to more settings, if needed. For example, counterfactually stable dynamics [2] can be used for non-deterministic discrete settings, while the monotone networks [3] can be used for continuous states and actions, as suggested by Lu et al 2020 [4]. For the benefit of the interested reader, we now briefly discuss these issues in Appendix F.
>
> As for graph structure: beyond assuming the arrow of time and Markovness, which tells us that the parent set of a time-t state is a subset of the states and actions at time t-1, we cannot make any claims about the identifiability of the causal graph topology (as it was assumed known rather than inferred). The revision has been updated to refer the interested reader to relevant works on structure discovery from the causal inference and RL literatures.
>
>
> **2. Comparison to approaches that learn factored masks for RL / Related Works**
>
> We have added a new Subsection 2.2 after the preliminaries that describe other related work, including many of the papers you referenced, and how they relate to MoCoDA. There is a large body of work here, as many have recognized the power of structured transitions for sample efficiency and generalization. It would be interesting to borrow or use some of the alternative approaches to incorporating structure in the dynamics model itself, as well as use one of the many approaches to unsupervised object detection to see if we can apply MoCoDA to pixel observations. To address your final question on more complicated evaluation scenarios here, we believe that as this ability to learn disentangled, entity-oriented representations improves, MoCoDA might become more broadly applicable, e.g., to the Procgen Benchmark or different Atari games. At the moment, this is not possible for lack of good representations. It remains to be seen how much of the advantage provided by data augmentation will already be captured by such a learned representation. But for CoDA and the works cited there-in (HER, RAD, etc.), we are not aware of other works that use structure to expand the training distribution for a model-free RL agent via data augmentation. We also built out the description of CoDA in preliminaries per your suggestion.
>
> References in part 2/2.

---

> > ### Author Response · Authors · 2022-08-01
> > **Response to Reviewer 8Yp5 (2/2)**
> >
> > **3. Long Horizon Structures**
> >
> > We agree that there are representations that will allow for disentanglement in long horizon relationships, and have added some language in Subsection 3.2 to reflect this.
> >
> > Regarding non-stationary graphs, we don’t consider time-varying graph structure as a matter of convenience, noting that we inherit from Pitis et al 2020 [5] the assumption that the local subspaces can be (approximately) inferred directly from the observed state and action at a given time index. Under this assumption, it is not necessary to consider which subspaces are likely to transition into which other subspaces, because we can always infer the subspace from the current observation. However, in a partially observable setting where this assumption is violated, we agree that such an approach may be required.
> >
> >
> > **References:**
> >
> > [1]. Pearl, Judea. Causality. Cambridge university press, 2009.
> >
> > [2] Oberst, Michael, and David Sontag. "Counterfactual off-policy evaluation with gumbel-max structural causal models." International Conference on Machine Learning. PMLR, 2019.
> >
> > [3] Lang, Bernhard. "Monotonic multi-layer perceptron networks as universal approximators." International conference on artificial neural networks. Springer, Berlin, Heidelberg, 2005.
> >
> > [4] Lu, Chaochao, et al. "Sample-efficient reinforcement learning via counterfactual-based data augmentation." NeurIPS workshop on Offline RL (2020).
> >
> > [5] Pitis, Silviu, Elliot Creager, and Animesh Garg. "[Counterfactual data augmentation using locally factored dynamics](https://arxiv.org/abs/2007.02863)." Advances in Neural Information Processing Systems 33 (2020): 3976-3990.

---

> > > ### Comment · Reviewer_8Yp5 · 2022-08-03
> > > **Feedback**
> > >
> > > Thank you very much for your detailed feedback. The responses and the updated main paper (especially those related to the causality) addressed my concerns about this work. I will raise my rating then.

---

### Official Review · Reviewer_FUqL · 2022-07-12

**Rating:** 7
**Confidence:** 3
**Soundness:** 3 good
**Presentation:** 3 good
**Contribution:** 2 fair

**Summary:**

This paper presents a novel model-based framework, MOCODA, as a generalization of the CODA framework for augmenting the training dataset of an offline-RL agent with samples generated using the parent distribution and locally factored dynamics, to solve zero-shot out-of-distribution (OOD) tasks. MOCODA is evaluated on continuous control tasks from the 2D Navigation and the HookSweep2 environments.

**Questions:**

- Why are there no CODA baselines? If I understand correctly, it makes sense to compare with CODA, right? And if not, it will be useful to have this addressed in the paper.
- One of the advantages of MOCODA is that it is applicable even when the parent sets overlap: are there cases where they do and does MOCODA perform well there?


I am happy to discuss and increase my score if these questions are answered.

**Limitations:**

Yes, the authors discuss the limitations of their work-- especially its potential inefficiency in training RL agents and suggest mitigating this by rebalancing MOCODA. The authors also discuss the societal impact of their work with respect to the handling of the causal assumptions made in this paper-- it would be good to see these listed out explicitly.

**Strengths And Weaknesses:**

*Strengths*

It's interesting to see the usage of locally factored dynamics models towards generalizing to unseen areas of the state space, and it does seem to reduce generalization error on some easy tasks.

*Weaknesses*

- It's not clear to me why the authors do not compare their method with CODA when they have presented SAC as an online RL model-free baseline.
- The authors say MOCODA works even when the parent sets overlap. Some analysis of the parent sets and the parent distribution would be useful, as it is not very clear right now why and how this is an advantage over CODA right now. (I have read B.2)
- I think the paper's clarity would greatly benefit from a clear pseudocode of their entire framework.
- The authors do not present any results to show the significance of using locally-factored models on the HookSweep2 tasks, which are considerably more difficult than the 2D Navigation tasks, and it's not clear why not.
- For improving the quality of the paper from the perspective of causal modeling, I think it would be helpful to have the causal assumptions made listed down specifically.

---

> ### Author Response · Authors · 2022-08-01
> **Response to Reviewer FUqL**
>
> Thank you for taking the time to review our paper and providing a detailed critique. We address your comments in three parts:
>
> **1. Regarding MoCoDA vs CoDA; Overlapping Parent Sets; Pseudocode**
>
> One of the main advantages of MoCoDA over CoDA [1] is that it works in case of overlapping parent sets. This is what allows us to use MoCoDA on the HookSweep2 task, where CoDA cannot** be applied.
>
> In order for CoDA to be applicable, it requires the transition graph to contain two or more completely disentangled connected components, so that they may be swapped—i.e. the parent sets must afford division into two non-overlapping groups. While this is the case in 3/4 of the 2D Navigation environment we used in our experiments, it is never the case for the main (Gripper, Object 1, Object 2) components in the HookSweep2 environment, precisely because it has overlapping parent sets. The parent sets for these components are:
>
> * (Gripper) → (Gripper)
> * (Gripper, Object 1) → (Object 1)
> * (Gripper, Object 2) → (Object 2)
>
> Because the Gripper is in the first three parent sets, we can’t directly apply CoDA to expand the support as shown in Figure 5.
>
> *There are two ways we might adapt CoDA in order to make it applicable to HookSweep2, but both are a bit complex.*
> - First, we might improve the heuristic mask we used in HookSweep2, to account for the independence between Gripper and Object when the Gripper is far from the Object. However, the state representation does not make this easy and the critical states involve interactions between Gripper and Objects, so we opted to treat the Gripper and Object as always entangled (our response to Reviewer 155Q explains why this is OK).
> - Second, it’s possible we could come up with a nearest-neighbors based conditional sampling scheme to extend the model-free CoDA algorithm in order to work with overlapping parent sets.  While this is arguably a large deviation from the CoDA method, we did experiment with this type of CoDA extension early on in this project, but were ultimately dissuaded by (a) the O(N_train* N_augment) time for data augmentation required for nearest-neighbors lookup and (b) numerical instabilities when applying fast k-nn libraries to OOD queries produced by the parent distribution model P_\theta(s, a).
>
> In the 2D Navigation experiment, CoDA’s requirement of disentangled causal mechanisms is satisfied in 3/4 of the state space, and so we were able to apply CoDA and rerun the Offline RL agents using the CoDA dataset. See updated Table 2 for results. Although CoDA does not perform as well as MoCoDA, it does manage to fill in the 2D Navigation environment, and the agents are able to learn near-optimal policies.
>
> We have included pseudocode for MoCoDA sampling as Algorithm 1 in Appendix E. For ease of exposition we provide pseudocode for the simplest variant of MoCoDA that assumes a single (global) FMDP (i.e. what is referred to in Table 1 as ‘MoCoDA’ with architecture ‘Globally Factored’), but discuss in Appendix E how this procedure can be easily extended to realize MoCoDA with locally factored dynamics.
>
> References:
>
> [1] Pitis, Silviu, Elliot Creager, and Animesh Garg. "Counterfactual data augmentation using locally factored dynamics." In NeurIPS 2020.
>
> **2. Regarding the significance of using locally-factored models on the HookSweep2 tasks.**
>
> We reran the HookSweep2 experiments, but generated the MoCoDA dataset using a fully connected (not factored) dynamics model that obtains comparable training loss (it is an ensemble of 5, 2-layer of 512 neuron MLP) to the locally factored model, instead of the locally factored dynamics model.
>
> We have included these ablation results in Table 3 of the paper. We see that while MoCoDA with a fully connected model still allows the agent to achieve some success in HookSweep2, it performs much worse than the MoCoDA with a locally-factored model. Further, we see that using the more OOD MoCoDA-P distribution does not help, suggesting that the fully connected model is failing to produce useful OOD transitions.
>
> **3. Regarding Causal Assumptions**
>
> We have updated the paper [Appendix F] to discuss how MoCoDA sampling relates to causal inference, including assumptions under which augmented samples drawn from a MoCoDA dynamics model can be interpreted as identifiable counterfactuals.

---

> ### Comment · Reviewer_FUqL · 2022-08-08
> **response to author rebuttal**
>
> Thanks to the authors for their rebuttal, the detailed clarifications, and the clearly outlined revisions.
>
> - the example provided of parent sets for the HookSweep2 environment makes the motivation behind MoCoDA much clearer for me, and it helps me understand why CoDA could not be applied to HookSweep2, so thanks! And I think it makes the exposition closer to asymptotic completeness to include the results on CoDA where applicable, as in the case of 2D navigation. It makes sense to see it there.
> - The MoCoDA pseudocode for the globally factored model separated out into the four core methods is helpful to keep a snapshot of the overall picture in mind.
> - Fully connected dynamics models do seem to perform much worse. It is interesting to see that in light of the authors' hypothesis that good models with "enough in-distribution data and the
>   right regularization might be able to respect local factorization".
> - Right, it makes sense then to talk of what is and is not meant by "counterfactual" in the paper since what the authors say they're doing is (model-based) "counterfactual" data augmentation. In the vein of being precise, I am not sure what is meant by saying MoCoDA does a "limited" form of Pearl-style counterfactual reasoning (in quotes)-- are there any other definitions of counterfactual reasoning? What is the simple answer to the question: what is the counterfactual distribution in your framework? Or, I guess, do you mean the interventional distribution? If so, what is the interventional distribution in your case? What does it mean to (not) consider counterfactuals over all the variables simultaneously-- since counterfactual questions are with respect to some specific set? Also, why use the terms causal inference and counterfactual reasoning interchangeably?-- The latter assumes the existence of an SCM to reason from whereas the former subsumes both reasoning and discovery/learning of the model. I know I am being a bit fussy about precise terminology and definitions but I think it's crucial given terms from the literature on causality are loosely used in deep learning texts and especially because it's central to your nomenclature.
>
> I am increasing my score from 5 to 7 given the additional experiments and clarifications, however, the lack of preciseness around the use of counterfactuals and the lack of analysis on model choice given it's a model-based method (though the authors comment on this as a possibility for future work) prevents me from further increasing it to 8.

---

> > ### Author Response · Authors · 2022-08-09
> > **alternate notions of "counterfactual"**
> >
> > We are glad to hear that the response helped to clarify some questions you had in the initial review, and we appreciate the increased score.
> >
> > Setting aside the scores and reviews, we definitely agree notions of causality and counterfactuals can be quite slippery at times, are used to mean a variety of things in different areas of the literature, and are worth being precise about. So we don’t mind the fussiness on these points. The feedback so far, from yourself and the other reviewers, has helped us considerably in improving the paper, and we will continue digesting these discussions as we work towards a final revision for the paper.
> >
> > There are indeed approaches to realizing “counterfactual” samples from different computational perspectives besides Pearl. For example the Rubin framework avoids the use of graphical models in favor of functional expressions fo “potential outcomes” of a treatment, denoted e.g. by $Y(T)$ with $T \in \{0, 1\}$ representing a binary treatment, perhaps [Rubin 2005]. These potential outcomes are sometimes colloquially called “counterfactuals”, despite Rubin’s stated preference against using this term [Rubin 2000]. These are closer to interventional quantities in a Pearlian setup, since there is no abduction/inference over exogenous noise sources prior to changing (intervening on) the treatment variable, although the Rubin framework can apply assumptions on the data generative process (e.g. no covariates downstream of the potential outcome) such that adding the abduction/inference stage wouldn’t change the computation.
> >
> > An example of the Rubin framework in ML is the application of propensity scores to offline bandit learning [Bottou et al 2013, Swaminathan and Joachims 2015]. Here the notion of “counterfactual” is in terms of off-policy evaluation and improvement.
> >
> > Perhaps the best way to understand MoCoDA is a lifting of the CoDA approach to data augmentation into a model-based setting, where we see the benefits in terms of handling overlapping parent sets. The distribution resulting from augmenting data in this way is closer to an interventional distribution, where the “intervention” in question involves setting time-t states and actions *for a specific causal mechanism* according to samples from the parent distribution $P_\theta(s,a)$ (indexing this sample to select the relevant dimensions) then feeding this through the forward dynamics $P_\phi(s’|s,a)$. This is not a Pearl-style counterfactual because we don’t condition on $s’$ to infer an exogenous noise variable before choosing $(s, a)$; rather, we simply choose $(s, a)$ as a sample from the parent distribution. In the rebuttal we have attempted to clarify that the MoCoDA models could be used to execute Pearl-style counterfactual sampling under some additional (admittedly restrictive) assumptions about the data generative process. Cf. the response to reviewer 8Yp5.
> >
> >
> > References:
> >
> > [Rubin 2005] *Causal Inference Using Potential Outcomes: Design, Modeling, Decisions*, Journal of the American Statistical Association, 2005
> >
> > [Rubin 2000] *Causal Inference without Counterfactuals: Comment*, Journal of the American Statistical Association, 2000
> >
> > [Bottou et al 2013] *Counterfactual Reasoning and Learning Systems: The Example of Computational Advertising.*, JMLR Vol. 14, 2013
> >
> > [Swaminathan and Joachims 2015] *Counterfactual Risk Minimization: Learning from Logged Bandit Feedback*, ICML 2015

---

### Official Review · Reviewer_155Q · 2022-07-15

**Rating:** 6
**Confidence:** 4
**Soundness:** 3 good
**Presentation:** 3 good
**Contribution:** 3 good

**Summary:**

This paper shows, both theoretically and experimentally, that a locally factored dynamics model improves sample efficiency and out-of-distribution generalisation when learning a dynamics transition model. In addition, this paper further proposes a Model-based Counterfactual Data Augmentation model to generate counterfactual transitions for RL, allowing offline RL agent to solve an out-of-distribution tasks.

**Questions:**

Please refer to the "Weakness"

**Limitations:**

Please refer to the "Weakness"

**Strengths And Weaknesses:**

Strengths:
1. The presentation is very clear and easy to follow.
2. Despite the fact that many existing works demonstrate that the known factored dynamics model can improve data efficiency, this paper may be the first to demonstrate that the factored dynamics model is also beneficial to out-of-distribution generalization.
3. The counterfactual data generation is interesting.
4. The experimental results clearly demonstrate the effectiveness of the proposed framework.

Weakness:
1. The framework assumes that the factored dynamics model is known. However, the factored model is unknown in most practical cases, and this paper does not propose solutions to this problem, limiting the framework's application scenarios.
2. This framework only evaluates a simple environment with four dim state spaces. It would be preferable to work in more complex environments, such as mujoco robot control tasks.

---

> ### Author Response · Authors · 2022-08-01
> **Response to Reviewer 155Q**
>
> Thank you for taking the time to review our paper and for your constructive commentary.
>
> We acknowledge that the current paper does not propose solutions for learning the dynamics factorization; we argue, however, that this is an open problem and somewhat complementary to our work. Several past and concurrent works aim to tackle unsupervised object detection [1, 2, 3, etc.] (i.e., learning an entity-oriented representation of states, which is a prerequisite for learning the dynamics factorization) and learning dynamics factorizations / causal models [4, 5, 6, etc.].
>
> Notably, neither problem is completely solved, but the cited works show that considerable progress is being made and gives us reason to believe that the proposed MoCoDA will become more widely applicable in the future as causal discovery progresses independently.
>
> We also note that in the meantime we could adopt an existing solution, e.g., the masked transformer, as leveraged in CoDA [6], to learn the local factorization in our experiments, we believe this would not add to our present contribution while potentially entangling empirical results.
>
> Further, we note that *the dynamics factorization used for MoCoDA does not need to be precisely known*: MoCoDA will work so long as the assumed graph structure contains the true structure (that is, it’s OK to have extraneous edges in any local causal structure). Thus, in our HookSweep2 experiment, where we did not actually know the true causal structure, we were able to use a heuristic structure based on physical distance that asserts independence between blocks whenever they are more than 5 centimeters apart in the simulation. Even though our chosen distance is not exact, we see that MoCoDA still works, because the heuristic structure contains the true causal structure. Indeed, the expanded training distribution of MoCoDA provides some benefit even when no causal structure is used (the dynamics are assumed fully connected), as shown in the new ablation in Table 3, in response to Reviewer FUqL.
>
> Re: only evaluating in a simple environment with four dim state spaces, we would like to point out that the HookSweep2 environment used is a Mujoco robot control task with 22 dimensions of observed state and 4 dimensions of action.
>
> [1] Dittadi, Andrea, Samuele Papa, Michele De Vita, Bernhard Schölkopf, Ole Winther, and Francesco Locatello. "Generalization and robustness implications in object-centric learning." In ICML 2022.
>
> [2] Lin, Zhixuan, Yi-Fu Wu, Skand Vishwanath Peri, Weihao Sun, Gautam Singh, Fei Deng, Jindong Jiang, and Sungjin Ahn. SPACE: Unsupervised Object Oriented Scene Representation via Spatial Attention and Decomposition. In ICLR 2020.
>
> [3] Locatello, Francesco, Dirk Weissenborn, Thomas Unterthiner, Aravindh Mahendran, Georg Heigold, Jakob Uszkoreit, Alexey Dosovitskiy, and Thomas Kipf.  Object-centric learning with slot attention. In NeurIPS 2020.
>
> [4] Kipf, Thomas, Elise van der Pol, and Max Welling. Contrastive learning of structured world models. In ICLR 2019.
>
> [5] Wang, Zizhao, Xuesu Xiao, Zifan Xu, Yuke Zhu, and Peter Stone. Causal Dynamics Learning for Task-Independent State Abstraction. In ICML 2022.
>
> [6] Pitis, Silviu, Elliot Creager, and Animesh Garg. "Counterfactual data augmentation using locally factored dynamics." In NeurIPS 2020.

---

### Official Review · Reviewer_1vMv · 2022-07-18

**Rating:** 7
**Confidence:** 3
**Soundness:** 3 good
**Presentation:** 3 good
**Contribution:** 3 good

**Summary:**

The approach in the paper is to learn a locally-factored dynamics model and the parent distribution of experience data, and use both to generate samples to augment experience data, towards helping the agent reduce sample complexity, and better generalize out of distribution. The paper also provides theoretical arguments justifying why this approach should work, an later considers a couple of tweaks on top of MoCoDa to address some limitations.

**Questions:**

- Lines 82-84 state an assumption that no two graphical models share the same structure. If the authors could spell out the implications (possible benefits and limitations) of this, and why this assumption is important, it would help readers better understand the framework.

**Limitations:**

Yes, this has been addressed adequately.

**Strengths And Weaknesses:**

This paper builds on the work of ref. [34], and provides interesting insights into the theoretical aspects of how a dynamics model helps generalize out-of-distribution with reduced sample complexity. Fig. 4 help understand how the factorized causal model helps overcome implicit data biases that might have otherwise hurt generalization. It is interesting to see how MoCoDa helps generalize beyond the empirical distribution, and across different architectures/algorithms.

The paper demonstrates an effective implementation of a simple/natural idea, is well-written, and should be of great interest to the audience at this conference.

[34]: Counterfactual data augmentation using locally factored dynamics. NeurIPS 2020

---

> ### Author Response · Authors · 2022-08-01
> **Response to Reviewer 1vMv**
>
> Thank you for your time and positive review.
>
> The assumption in lines 82-84 is a simplifying assumption that allows us to avoid some complexity that would arise from local subspaces that share causal structures but have different dynamics.
>
> An example of this could be an arena that doubles as a hockey rink in the winter and a basketball court in the summer. The causal graph for any particular scene taking place in the arena would be the same regardless of whether the ground is ice, but the dynamics would be different. While our current treatment does not handle this case (which makes counting local subspaces in our theory slightly easier, and simplifies our implementation of the locally factored dynamics model), we might accommodate it by identifying local subspaces using a latent variable rather than the mask itself.
>
> Per your suggestion, we have now included a note in the main text.

---

### Author Response · Authors · 2022-08-01
**General response**

We are grateful to all the reviewers for their detailed and insightful feedback on our submission.

To briefly summarize the reviews: all four reviewers tended towards recommending acceptance [1vMv,155Q,FUqL,8Yp5]. Reviewers reported that our analysis sheds light on how locally factored dynamics improve OOD generalization and sample complexity [1vMv,155Q,8Yp5], and that our proposed algorithm MoCoDA generated data that helped downstream agents generalize beyond the available experience to improve performance on novel OOD tasks [1vMv,155Q,FUqL,8Yp5].

However, a few questions emerged, which we list below:
* Can we clarify our assumptions about the data generative process [1vMv,FUqL,8Yp5]? Reviewers were especially interested in causal assumptions [FUqL,8Yp5], the question of identifiability of the causal structure [155Q,8Yp5], and identifiability of augmented transitions sampled from the learned dynamics model $P_\theta(s,a)P_\phi(s’|s,a)$ [8Yp5].
* Can we discuss the conceptual distinctions [FUqL,8Yp5] between our proposed method MoCoDA and CoDA [Pitis et al 2020]?
* Can we show, in further detail, the empirical benefit of using MoCoDA by adding targeted baselines to the experiments? Specifically, we were asked to compare against CoDA [FUqL] and an ablated version of MoCoDA where a dense transition model is used in place of a locally factored transition model [FUqL].
* Can we discuss relevant related works that seek to improve RL performance via learning a structured dynamics model [8Yp5]?

In our rebuttal, we answer all of these questions in the affirmative, while discussing various nuances that arise. The rebuttal has been organized in terms of individual responses to each reviewer. We have also submitted a pdf revision, with new additions in Green. Should any additional questions arise, or if further clarification is needed, we will monitor this page throughout the discussion period.

References:

[Pitis et al 2020] Pitis, Silviu, Elliot Creager, and Animesh Garg. "[Counterfactual data augmentation using locally factored dynamics](https://arxiv.org/abs/2007.02863)." Advances in Neural Information Processing Systems 33 (2020): 3976-3990.

---

### Meta-Review · Area_Chair_XGXi · 2022-08-29

**Recommendation:** Accept
**Confidence:** Certain

**Metareview:**

The paper suggests to improve sample efficiency and out-of-distribution generalization in RL by learning locally factored world models, and use these models to generate counterfactual data to train on. The key assumption is that the environment model is the right model to factorize (as opposed to, say a policy or value function) and that this model will generalize out of distribution when performing the relevant interventions. All reviewers were in agreement the paper was well written and presented an interesting idea with sound empirical verification. Several comments pointed to an unclear definition of 'counterfactual' used in the paper (as the authors point out, it means different things depending on whether adopting a potential outcome or DAG framework) - please make sure this is clear in the final version, as well as a clear explanation of the distinction with CODA.

**Award:**

No

---

### Decision · Program_Chairs · 2022-09-14

Accept